# Biodegradable black phosphorus-based nanospheres for *in vivo* photothermal cancer therapy

Jundong Shao[1,2], Hanhan Xie[1], Hao Huang[1], Zhibin Li[1], Zhengbo Sun[2], Yanhua Xu[2], Quanlan Xiao[2], Xue-Feng Yu[1], Yuetao Zhao[1], Han Zhang[2], Huaiyu Wang[1] & Paul K. Chu[3]

Photothermal therapy (PTT) offers many advantages such as high efficiency and minimal invasiveness, but clinical adoption of PTT nanoagents have been stifled by unresolved concerns such as the biodegradability as well as long-term toxicity. Herein, poly (lactic-co-glycolic acid) (PLGA) loaded with black phosphorus quantum dots (BPQDs) is processed by an emulsion method to produce biodegradable BPQDs/PLGA nanospheres. The hydrophobic PLGA not only isolates the interior BPQDs from oxygen and water to enhance the photothermal stability, but also control the degradation rate of the BPQDs. The *in vitro* and *in vivo* experiments demonstrate that the BPQDs/PLGA nanospheres have inappreciable toxicity and good biocompatibility, and possess excellent PTT efficiency and tumour targeting ability as evidenced by highly efficient tumour ablation under near infrared (NIR) laser illumination. These BP-based nanospheres combine biodegradability and biocompatibility with high PTT efficiency, thus promising high clinical potential.

[1] Institute of Biomedicine and Biotechnology, Shenzhen Institutes of Advanced Technology, Chinese Academy of Sciences, Shenzhen 518055, China. [2] SZU-NUS Collaborative Innovation Center for Optoelectronic Science and Technology, and Key Laboratory of Optoelectronic Devices and Systems of Ministry of Education and Guangdong Province, College of Optoelectronic Engineering, Shenzhen University, Shenzhen 518060, China. [3] Department of Physics and Materials Science, City University of Hong Kong, Tat Chee Avenue, Kowloon, Hong Kong 999077, China. Correspondence and requests for materials should be addressed to X.F.Y. (email: xf.yu@siat.ac.cn) or to H.Z. (email:hzhang@szu.edu.cn) or to P.K.C. (email:paul.chu@cityu.edu.hk).

Development of novel nanomaterials and advanced nanotechnology for cancer treatment has attracted much interest[1]. As a promising alternative or supplement to traditional cancer therapy, photothermal therapy (PTT) based on the interaction between tissues and near infrared (NIR) radiation offers many advantages such as high efficiency and minimal invasiveness[2–5]. Owing to the excellent NIR optical performance, nanomaterials such as metallic nanostructures, metal-based semiconductor nanoparticles and carbon nanomaterials have been explored and employed as PTT agents or drug release systems in *in vivo* cancer therapy[6–15]. Nanomaterials with a size range between 20 and 200 nm circumvent rapid renal filtration enabling passive accumulation in tumours at high concentrations for a longer time than organic molecules via the enhanced permeability and retention (EPR) effect that can hardly be achieved by other molecular agents[16–19]. However, unlike other small biodegradable molecules, inorganic nanoparticles generally have poor biodegradability and stay in the body for a long period of time accentuating the risk of deleterious effects. Hence, clinical adoption of nanomaterials hinges on the proper control of biodegradability as well as long-term toxicity of the materials and by-products[20,21]. It has recently been reported that ultrasmall nanoparticles (<10 nm) undergo rapid renal filtration[22] but suffer from a short blood circulation half-life, attenuated EPR effects, as well as reduced tumour accumulation and retention. Therefore, it is highly desirable to develop new PTT agents which have not only the proper size enabling efficient tumour targeting, but also good biocompatibility and biodegradability ensuring that the nanoparticles can be discharged harmlessly from the body in a reasonable period of time after completion of the designed therapeutic functions.

As a new member of 2D materials, atomically thin black phosphorus (BP) has many potential applications in electronics and optoelectronics[23–31]. Being a metal-free layered semiconductor, BP has a layer-dependent direct bandgap varying from 0.3 eV for the bulk materials to 2.0 eV for phosphorene (monolayered BP), thereby allowing broad absorption across the ultraviolet and infrared regions[25]. Liquid exfoliation methods are commonly utilized to prepare BP nanosheets with different thicknesses and sizes[32–36] for bioimaging and phototherapy[37–39]. In particular, ultrasmall BP nanosheets (also called BP quantum dots, BPQDs) with a size of ~3 nm have a large NIR extinction coefficient, high photothermal conversion efficiency and little cytotoxicity[39]. As an inorganic nanoagent, BP is attractive due to its inherent biocompatibility and furthermore, as one of the vital elements, *P* is a benign element making up ~1% of the total body weight as a bone constituent in the human body[40–42]. Recent experiments have shown that BP nanosheets, especially ones with a small thickness and size, have high reactivity with oxygen and water[43–46] and can degrade in aqueous media. Moreover, the final degradation products are nontoxic phosphate and phosphonate[45–47], both of which exist in and are well tolerated by the human body[41,42]. Therefore, ultrasmall BPQDs with good photothermal performance and biocompatibility are potential therapeutic agents. However, their actual clinical application *in vivo* still suffers from rapid renal excretion and degradation of the optical properties during circulation in the body.

In this work, to accomplish high therapeutic efficacy and desirable biodegradation, poly (lactic-co-glycolic acid) (PLGA) loaded with 3 nm BPQDs is processed by an oil-in-water emulsion solvent evaporation method to produce ~100 nm BPQDs/PLGA nanospheres (NSs). PLGA, a well known biodegradable and biocompatible polymer approved by the Food and Drug Administration (FDA), is widely used as a vehicle in the delivery of drugs and nanomaterials[48–51]. In general, the degradation period of PLGA spans several months and its degradation rate can be controlled by adjusting the chemical composition such as the monomer ratio and molecular weight[52]. Herein, the hydrophobic PLGA not only isolates the interior BPQDs from oxygen and water to enhance the photothermal stability, but also controls the degradation rate of the BPQDs in the physiological medium. The BPQDs/PLGA NSs with the optimal size enable prolonged blood circulation and effective accumulation in tumours based on the EPR effect. Furthermore, *in vitro* and *in vivo* experiments are performed systematically to evaluate the performance of the BPQDs/PLGA NSs as biodegradable and biocompatible PTT agents in cancer therapy.

## Results

**Morphology and characterization.** The ultrasmall BPQDs were prepared by a modified liquid exfoliation technique according to the method reported by our group previously[39]. The transmission electron microscopy (TEM) image in Fig. 1a reveals the uniform morphology of the BPQDs and the high-resolution TEM image in Fig. 1b discloses lattice fringes of 0.34 nm ascribed to the (021) plane of the BP crystal[33]. The average lateral size of $3.1 \pm 1.8$ nm (Fig. 1c) is obtained according to the statistical TEM analysis of 200 BPQDs. The atomic force microscopy (AFM) image in Fig. 1d shows the topographic morphology of the BPQDs and the thickness is determined by the cross-sectional analysis. The measured heights of 2.4, $\sim 1.8$ and 1.3 nm (Fig. 1e) correspond to BPQDs with $\sim 4$, 3 and 2 quintuple layers, respectively. Statistical analysis of the AFM data from 200 BPQDs yields an average thickness of $1.8 \pm 0.6$ nm corresponding to a stack of $3 \pm 1$ quintuple layers of BP.

The BPQDs/PLGA NSs were synthesized by an oil-in-water emulsion solvent evaporation method. As shown in Fig. 1f, the scanning electron microscopy (SEM) image provides evidence of high-yield synthesis of the BPQDs/PLGA NSs. The high-magnification SEM image in Fig. 1g confirms the uniform spherical shape with a smooth surface and the statistical analysis (Fig. 1h) of 200 NSs discloses an average size of $102.8 \pm 35.7$ nm. The average hydrodynamic size of the BPQDs/PLGA NSs is $\sim 127.6 \pm 43.8$ nm (Supplementary Fig. 1), which is slightly larger than that determined by SEM and hence, the size of the BPQDs/PLGA NSs is within the accepted range enabling efficient uptake by tumours based on the EPR effect[53,54]. The TEM images in Fig. 1i and inset depict the internal structure of the BPQDs/PLGA NSs. Many BPQDs are incorporated into each NS and most of them are located inside and protected by the PLGA shells. The loading efficiency and encapsulation efficiency are 12.9 and 83.8% as determined by energy dispersive X-ray spectroscopy (Fig. 1j) and inductively coupled plasma atomic emission spectroscopy. Raman scattering is further performed to characterize the BPQDs and BPQDs/PLGA NSs (Supplementary Fig. 2). Both samples exhibit three prominent Raman peaks at 359.7, 436.2 and 463.5 cm$^{-1}$ which can be assigned to one out-of-plane phonon mode ($A^1_g$) and two in-plane modes ($B_{2g}$ and $A^2_g$) of BP, respectively, confirming the introduction of BPQDs into PLGA NSs.

**Stability evaluation under ambient conditions.** To evaluate the influence of PLGA encapsulation on the BPQDs stablity, the BPQDs and BPQDs/PLGA NSs with the same amount of BPQDs (20 p.p.m.) were dispersed in water and exposed to air for 8 days and then their optical properties were examined at predetermined time intervals (0, 2, 4, 6 and 8 days). As shown in Fig. 2a, the colour of the solution containing the bare BPQDs become lighter during dispersion, while the BPQDs/PLGA NSs solution retains the colour quite well. The inset photographs in Fig. 2b,c show that both dispersions are stable without visible aggregation or

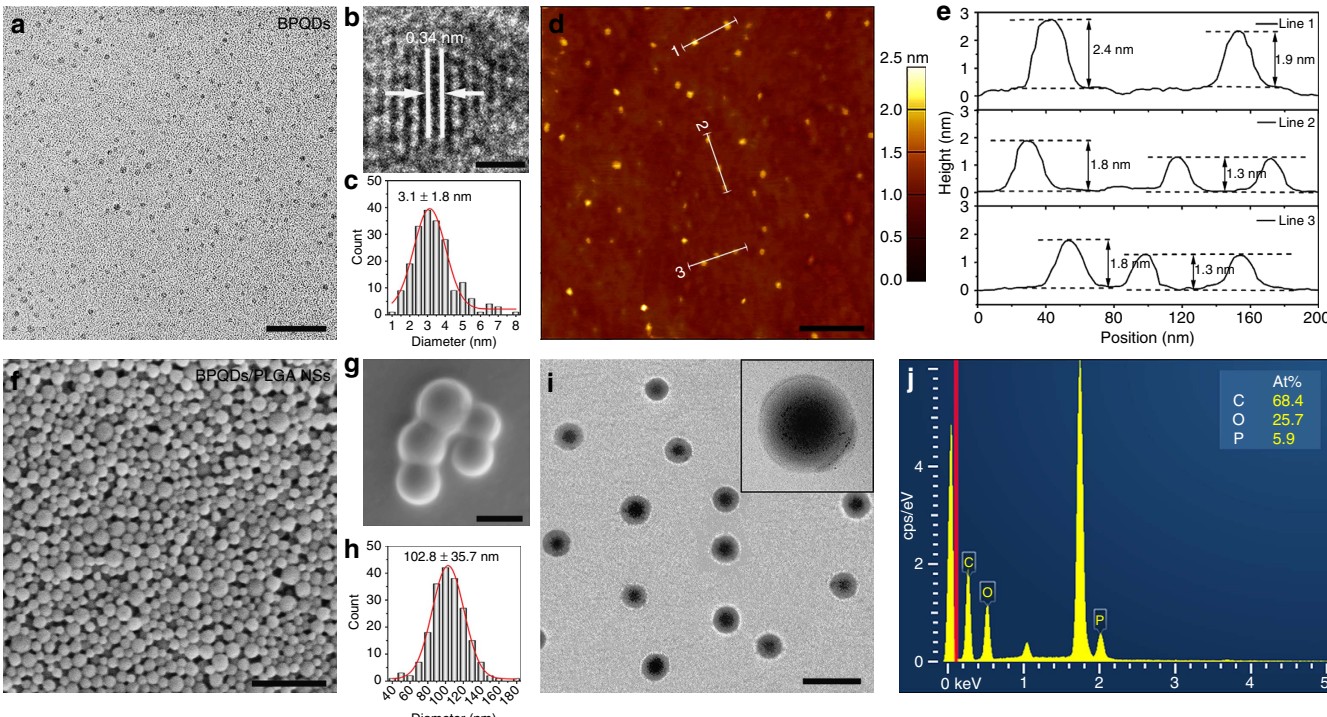

**Figure 1 | Morphology and characterization.** (**a**) TEM (scale bar, 20 nm) and (**b**) high-resolution TEM images of the BPQDs (scale bar, 1 nm). (**c**) Statistical analysis of the size of 200 BPQDs based on the TEM images. (**d**) AFM image of the BPQDs (scale bar, 200 nm). (**e**) Height profiles along the white lines in **d**. (**f,g**) SEM images of the BPQDs/PLGA NSs (scale bar, **f**: 1 μm; **g**: 100 nm). (**h**) Statistical analysis of the size of 200 BPQDs/PLGA NSs according to the SEM images. (**i**) TEM image of the BPQDs/PLGA NSs (scale bar, 200 nm) with the inset displaying the magnified TEM image of a BPQDs/PLGA NS. (**j**) Energy dispersive X-ray spectroscopy analysis of the BPQDs/PLGA NSs.

sedimentation and it can be confirmed by the Tyndall effect for diluted suspensions. The corresponding absorption spectra are displayed in Fig. 2b,c. During initial dispersion, both the BPQDs and BPQDs/PLGA NSs exhibit a typical broad absorption band spanning the ultraviolet and NIR regions. However, the absorbance intensity of the BPQDs in water decreases with dispersion time. The inset spectra in Fig. 2b show the variation in the absorption ratios at 808 nm and the intensity ($A$) decreases by 27.5% compared with the original value ($A_0$) after 2 days and 62.5% after 8 days. In contrast, the absorbance of the BPQDs/PLGA NSs is more stable as shown in Fig. 2c and the absorbance intensity only drops by 9.7% after 8 days. Degradation of BP in water is caused by the irreversible reaction with oxygen and water forming oxidized phosphorus species ($P \rightarrow P_xO_y$) followed by conversion of $P_xO_y$ to the final anions (that is, $PO_4^{3-}$)[45]. Hence, the absorbance decreases because of the degradation of the bare BPQDs. In contrast, the BPQDs/PLGA NSs show stronger absorption than the BPQDs due to the contribution of the PLGA shells and the larger size of the whole particle. Owing to the strong hydrophobicity of PLGA, the BPQDs are adequately protected from oxygen and water leading to the enhanced stability of the BPQDs/PLGA NSs in water under ambient conditions. Furthermore, comparing with obvious blue-shift of the Raman spectra of the BPQDs after dispersion in water for 8 days (Fig. 2d), no significant change is observed from that of the BPQDs/PLGA NSs (Fig. 2e), confirming their better stability[55,56].

The stability of the photothermal performance of the BPQDs and BPQDs/PLGA NSs in water is further examined (Fig. 2f,g). In the beginning, the temperature of the BPQDs solution increases by 19.3 °C after 808 nm laser irradiation (1.0 W cm$^{-2}$) for 10 min but after 8 days, the temperature rise is only 8.7 °C due to the BP degradation. In comparison, after 8 days, the temperature of the

BPQDs/PLGA NSs solution increases by 19.9 °C after irradiation for 10 min and it is very close to the initial one of 21.1 °C. It indicates that PLGA encapsulation can effectively prevent the degradation of BPQDs and maintain the photothermal characteristics in water. However, ∼10% loss in the absorbance and photothermal performance is observed from the BPQDs/PLGA NSs after dispersion in water for 8 days due to the biodegradable nature of PLGA in water[48–51] consequently exposing a small amount of the BPQDs to water. Nonetheless, since PLGA almost does not degrade in air, the BPQDs/PLGA NSs can be stored as powders for at least 3 months (Supplementary Fig. 3).

**Biodegradation behaviour of the BPQDs and BPQDs/PLGA NSs.** The biodegradation behaviour of the BPQDs and BPQDs/PLGA NSs in the phosphate-buffered saline (PBS; pH 7.4) is investigated in a horizontal shaker at 37 °C. PBS is a suitable medium to study biodegradation *in vitro*[57]. Compared with natural dispersion in water, the bare BPQDs in PBS subjected to agitation degrade faster giving rise to greater absorption and photothermal loss during 24 h (Supplementary Fig. 4). In contrast, the BPQDs/PLGA NSs maintain the stability and performance during the initial 24 h (Fig. 3a,b).

The long-term biodegradability of the BPQDs/PLGA NSs in PBS is further assessed. As shown in Fig. 3a,b, the absorption and photothermal performance of the BPQDs/PLGA NSs deteriorates after 8 weeks due to the biodegradability of PLGA. The residual weights of the BPQDs/PLGA NSs during degradation are measured and the residual weight percentage is determined by the following formula: residual weight (%) = $Wr \times 100 / Wo$, where $Wr$ is the dry weight of the sample after degradation and

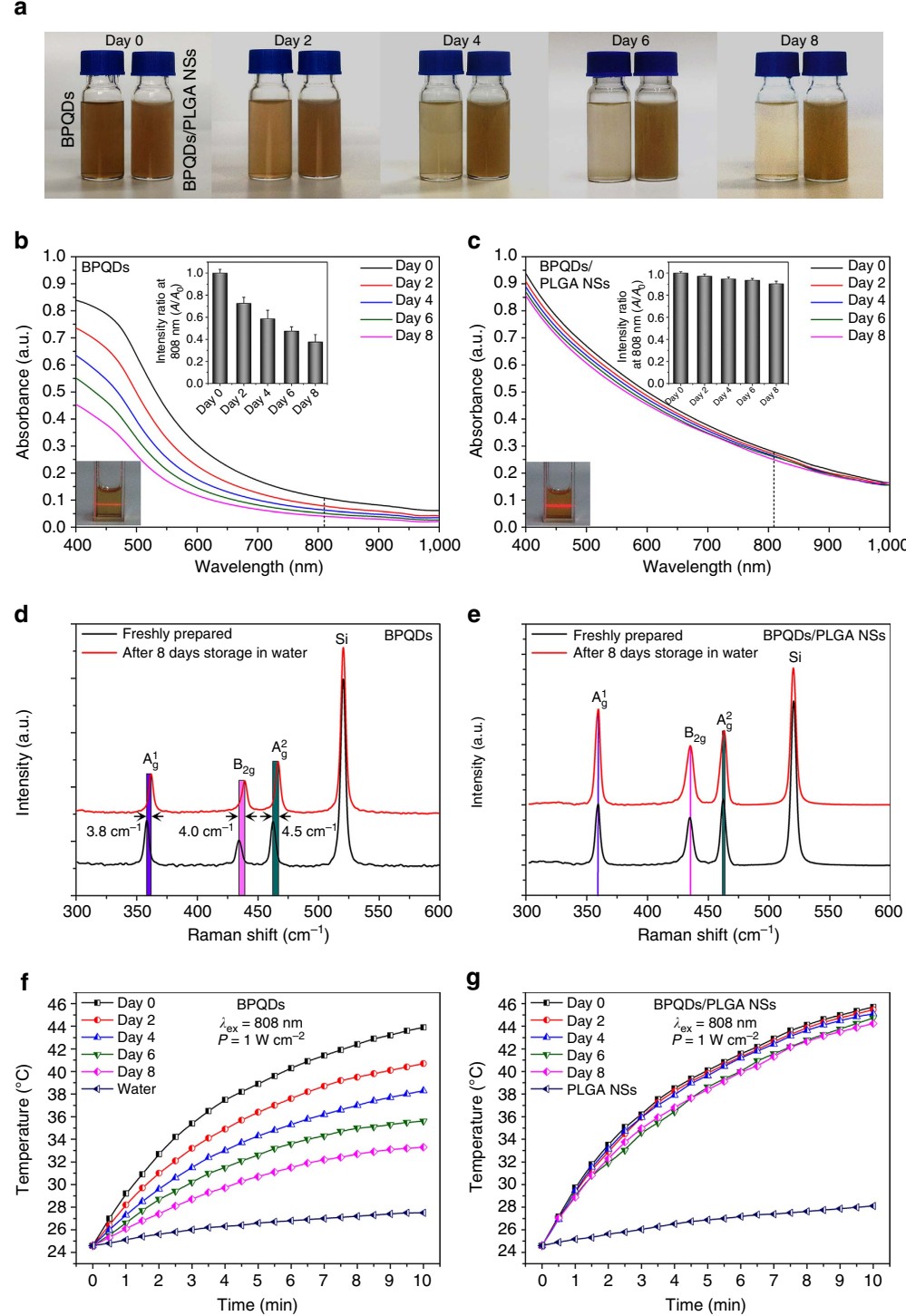

**Figure 2 | Stability evaluation under ambient conditions.** (**a**) Photographs and (**b,c**) Absorption spectra of the BPQDs and BPQDs/PLGA NSs with the same amount of BPQDs (20 p.p.m.) after storing in water for different periods of time. Insets in (**b,c**): tyndall effect and variation of the absorption ratios ($A/A_0$) at 808 nm. (**d,e**) Raman scattering spectra acquired from the BPQDs and BPQDs/PLGA NSs, respectively, after storing in water for 0 and 8 days. (**f,g**) Photothermal heating curves of the BPQDs and BPQDs/PLGA NSs, respectively, after storing in water for different periods of time and being irradiated with the 808 nm laser (1 W cm$^{-2}$) for 10 min.

$Wo$ is the initial weight of the sample. As shown in Fig. 3c, the residual weight of the BPQDs/PLGA NSs exhibits a downward trend with a small degradation rate during the first week but it accelerates showing nearly 80% loss after 8 weeks. Figure 3d shows the corresponding SEM and TEM images. After 1 week, the NSs maintain the integrity generally with only slight morphological changes (arrow indicated). After 4 weeks,

degradation is visible as indicated by the shape change and after 8 weeks, the morphology of the NSs is completely disrupted and very few residues of the BPQDs can be observed (arrow indicated). The influence of laser irradiation on the degradation of the BPQDs/PLGA NSs is further assessed (Supplementary Fig. 5). After laser illumination (808 nm, 1 W cm$^{-2}$) for 10 min, no evident morphological change and influence on the

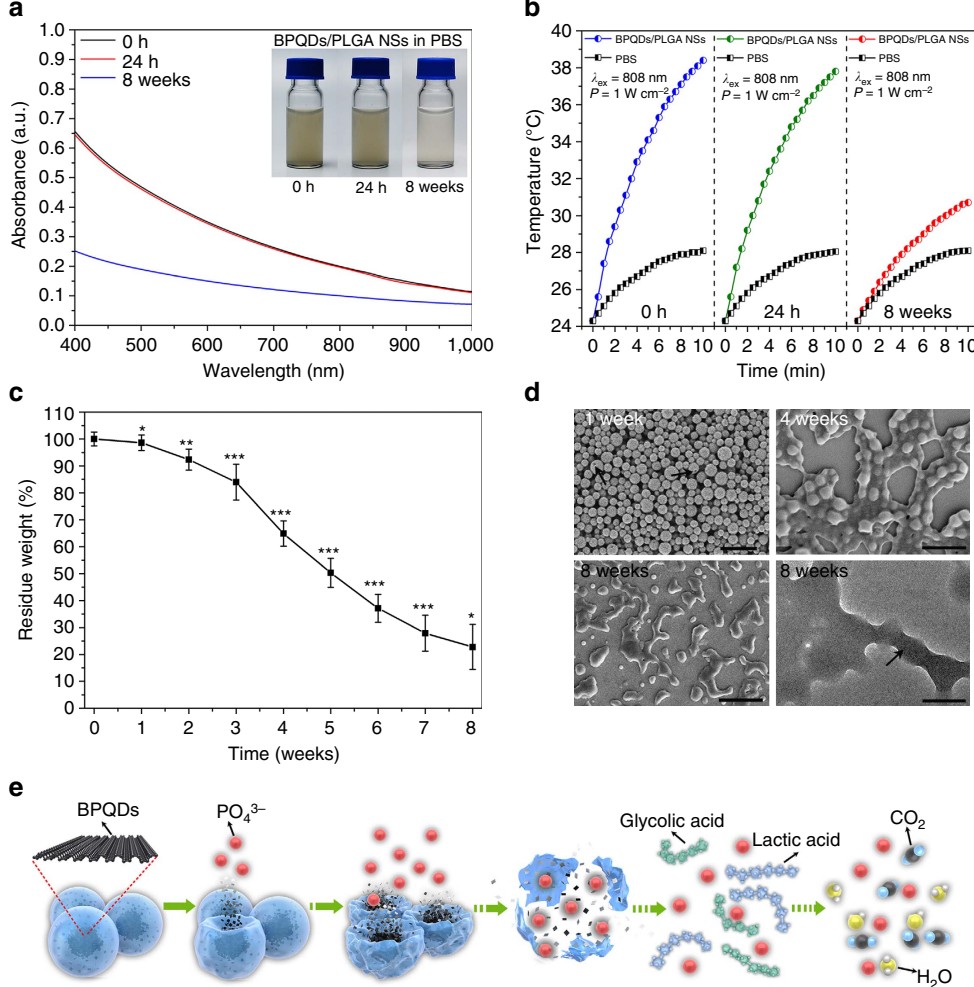

**Figure 3 | Biodegradation performance. (a)** Absorbance spectra of the BPQDs/PLGA NSs (internal BPQDs concentration is 10 p.p.m.) dispersed in PBS for 0 h, 24 h and 8 weeks with the inset showing the corresponding photographs. **(b)** Photothermal heating curves of the BPQDs/PLGA NSs dispersed in PBS for 0 h, 24 h and 8 weeks and irradiated with the 808 nm laser (1 W cm$^{-2}$) for 10 min. **(c)** Residual weight of the BPQDs/PLGA NSs after degradation in PBS as a function of time ($n = 5$; $^{*}P < 0.05$, $^{**}P < 0.01$, $^{***}P < 0.001$; ANOVA). **(d)** SEM images (scale bars, 500 nm) of the BPQDs/PLGA NSs after degradation in PBS for 1, 4 and 8 weeks together with the corresponding TEM image (scale bar, 200 nm) of the NSs after degradation for 8 weeks. **(e)** Schematic representation of the degradation process of the BPQDs/PLGA NSs in the physiological environment.

degradation rate can be found from the NSs suggesting that the photothermal treatment does not affect the biodegradability of the NSs significantly.

The degradation process is illustrated in Fig. 3e. When the BPQDs/PLGA NSs are in the physiological environment, the external PLGA shells degrade gradually due to hydrolysis of the ester linkage into segments (reduced molecular weight), oligomers and monomers, and finally carbon dioxide and water[52,58]. Degradation of PLGA disrupts the NSs and triggers release of the interior BPQDs which degrade rapidly if they are not protected by PLGA. The final degradation products from the BPQDs are nontoxic phosphate and phosphonate[45–47], both of which are commonly found in the human body[41,42]. In *in vivo* applications, the unique biodegradability of the BPQDs/PLGA NSs not only circumvents rapid degradation of the optical performance, but also enables harmless clearance from the body in a reasonable period of time after fulfilling their therapeutic functions.

***In vitro* cytotoxicity assays and photothermal experiments.** The possible cytotoxicity of the BPQDs/PLGA NSs towards normal

and tumour cells is probed (Fig. 4a). The relative viability of the human skin fibroblast, MCF7 (human breast cancer cells) and B16 (melanoma cells) cells incubated with the BPQDs/PLGA NSs (internal BPQDs concentration is 0, 2, 5, 10, 20, 50 and 100 p.p.m.) for 48 h was determined by the cell counting kit-8 (CCK-8) assay. No significant cytotoxicity can be observed from all types of cells even at the concentration of BPQDs as high as 100 p.p.m., which far surpasses that used in the following photothermal experiments.

The photothermal ability of the BPQDs/PLGA NSs for killing cancer cells is further investigated. After incubation with the NSs for 4 h, the MCF7 and B16 cells were irradiated with the 808 nm laser (1 W cm$^{-2}$) for 10 min and the cell viability was quantitatively assessed by the CCK-8 assay (Fig. 4b). A dose-dependent PTT effect can be observed. Almost all of the cancer cells are killed after incubating with the NSs containing only 10 p.p.m. of BPQDs and exposure to the NIR laser but on the contrary, direct irradiation of the cells in the absence of the NSs does not compromise the cell viability. Similar results can be observed from the fluorescence micrographs (Fig. 4c) of the cells co-stained by calcein AM (live cells, green fluorescence) and propidium iodide (PI, dead cells, red fluorescence) after the PTT

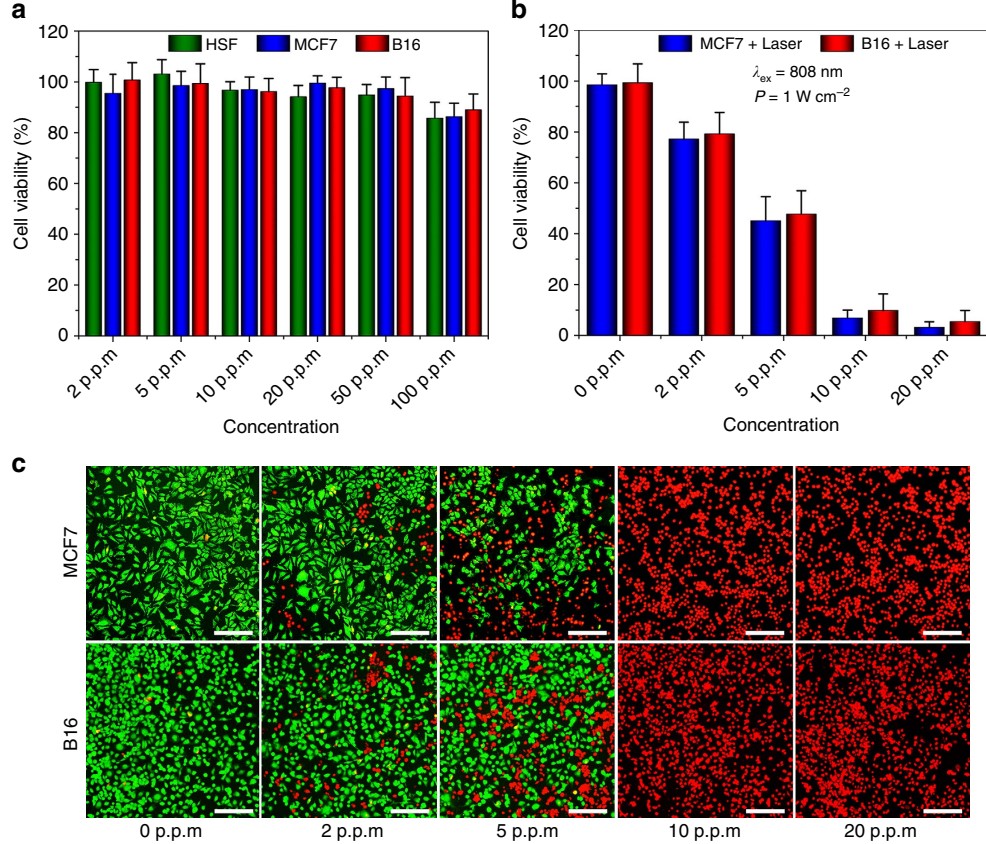

**Figure 4 | Cell experiments.** (**a**) Relative viability of the human skin fibroblast normal cells, MCF7 cancer cells and B16 melanoma cells after incubation with BPQDs/PLGA NSs (internal BPQDs concentrations of 0, 2, 5, 10, 20, 50 and 100 p.p.m.) for 48 h. (**b**) Relative viability of the MCF7 and B16 cells after incubation with BPQDs/PLGA NSs (internal BPQDs concentrations of 0, 2, 5, 10 and 20 p.p.m.) for 4 h after irradiation with the 808 nm laser (1 W cm$^{-2}$) for 10 min. (**c**) Corresponding fluorescence images (scale bars, 100 μm for all panels) of the cells stained with calcein AM (live cells, green fluorescence) and PI (dead cells, red fluorescence).

treatment. The results demonstrate the good PTT efficiency of the BPQDs/PLGA NSs in sterilizing cancer cells.

In the next step, the PTT efficiency of the BPQDs/PLGA NSs was compared with that of gold nanorods (AuNRs), one of the common photothermal agents. On account of the large NIR extinction coefficient and high photothermal conversion efficiency of the BPQDs (ref. 39), the BPQDs/PLGA NSs are more efficient in increasing the solution temperature than the AuNRs (Supplementary Fig. 6). In the cell photothermal experiments, both of the NSs (containing only 10 p.p.m. of BPQDs) and AuNRs (72.4 p.p.m.) can kill the cancer cells almost completely, but it is clear that less BPQDs are needed. These results confirm the suitability of BPQDs/PLGA NSs as an efficient PTT agent.

***In vivo* toxicity.** The *in vivo* toxicology of the BPQDs/PLGA NSs is investigated systematically. Sixty healthy female Balb/c mice (6 weeks old) were randomly divided into 4 groups and subjected to variable conditions, including: (1) control group without any treatment, (2) NSs directly intravenously injected into the mice, (3) NSs intravenously injected into the mice after 808 nm laser irradiation for 10 min and (4) NSs intravenously injected into the mice which are then exposed to artificial daylight for 24 h. The injection dose of the NSs is ∼10 mg BP kg$^{-1}$ and haemato-logical, blood biochemical and histological analyses were performed at time points of 1, 7 and 28 days post-injection.

The standard haematology markers including the white blood cells, red blood cells, haemoglobin, mean corpuscular volume, mean corpuscular haemoglobin, mean corpuscular haemoglobin concentration, platelets and haematocrit were measured (Fig. 5a). Compared with the control group, all the parameters in the three NSs-treated groups at all time points appear to be normal and the differences between are not statistically significant (*P* value > 0.05). These results indicate that the BPQDs/PLGA NSs do not cause obvious infection and inflammation in the treated mice[59].

Blood biochemical analyses were carried out and various parameters including alanine transaminase, aspartate transaminase, total protein, globulin, total bilirubin, blood urea nitrogen, creatinine and albumin were examined (Fig. 5b). Compared with the control group, no meaningful difference can be observed from the three NSs-treated groups at all time points. Hence, the NSs treatment does not affect the blood chemistry of mice. Further-more, since alanine transaminase, aspartate transaminase and creatinine are closely related to the functions of the liver and kidney of mice[59], the results demonstrate that the NSs induce no obvious hepatic and kidney toxicity in mice.

Finally, the corresponding histological changes of organs were checked by immunohistochemistry using major organs including the liver, spleen, kidney, heart and lung collected and sliced for haematoxylin and eosin staining (Fig. 5c). No noticeable signal of organ damage can be observed during the whole-treatment period from all the groups suggesting no apparent histological

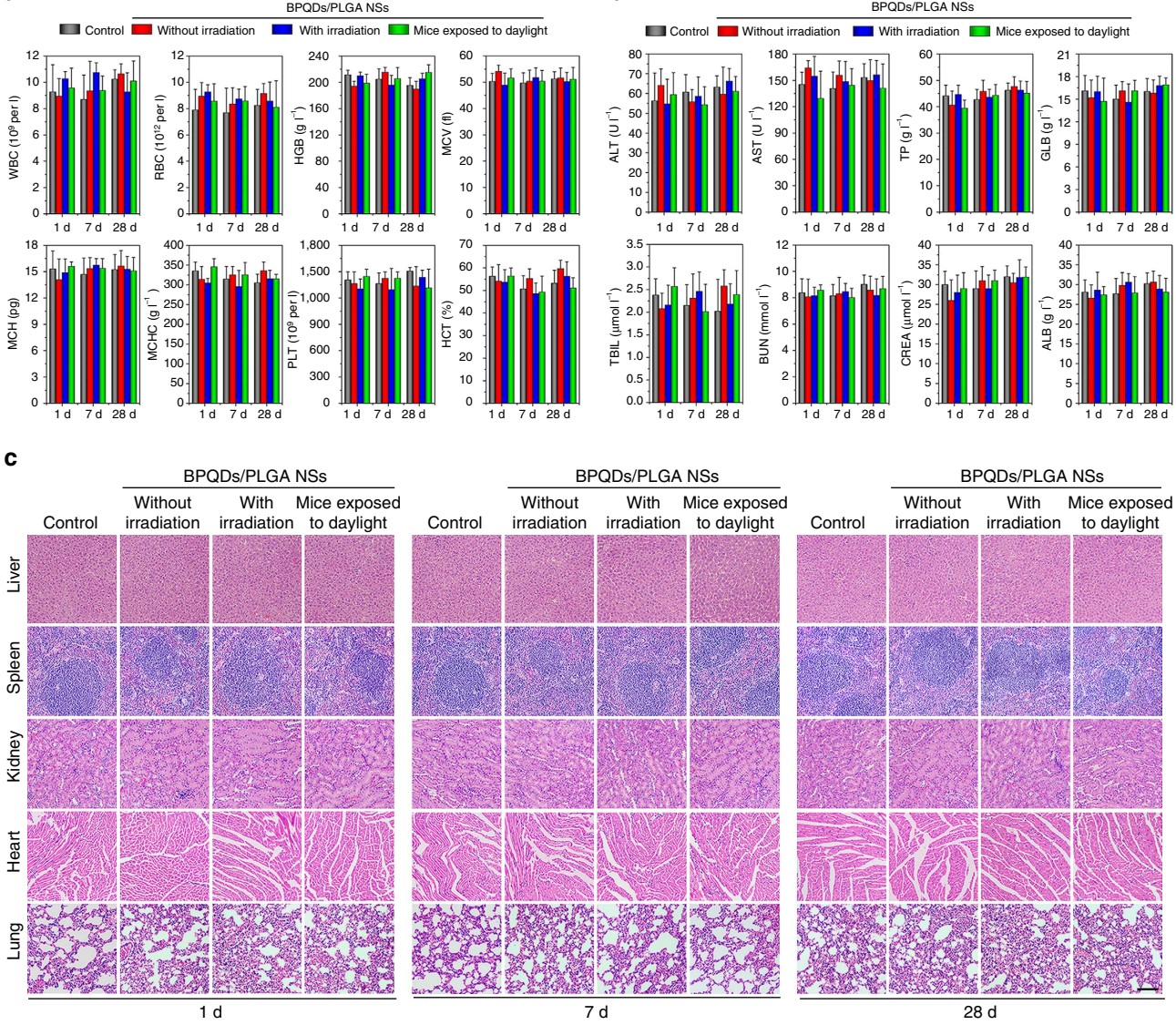

**Figure 5 | *In vivo* toxicity.** (**a**) Haematological data of the mice intravenously injected with the BPQDs/PLGA NSs at 1, 7 and 28 days post-injection. The terms are following: white blood cells, red blood cells, haemoglobin, mean corpuscular volume, mean corpuscular haemoglobin, mean corpuscular haemoglobin concentration, platelets and haematocrit. (**b**) Blood biochemical analysis of the NSs-treated mice at 1, 7 and 28 days post-injection. The results show the mean and s.d. of aminotransferase, aminotransferase, total protein, globulin, total bilirubin, blood urea nitrogen, creatinine and albumin (AL B). (**c**) Histological data (haematoxylin and eosin stained images) obtained from the liver, spleen, kidney, heart and lung of the NSs-treated mice at 1, 7 and 28 days post-injection (scale bars, 100 μm for all panels).

abnormalities or lesions in the NSs-treated groups for the test dose.

According to above analyses, inappreciable toxicity is observed from the BPQDs/PLGA NSs regardless of NIR laser irradiation. Even if the NSs-treated mice are under artificial daylight illumination for 24 h, no significant toxic side effects can be found, indicating that the NSs induce no evident phototoxicity which has generally been observed from many photosensitizer molecules[60]. These results demonstrate the good biocompatibility of the BPQDs/PLGA NSs.

***In vivo* biodistribution.** Since significant amounts of *P* exist in the animal body, it is very difficult to direct obtain biodistribution information of the BP-based materials (Supplementary Fig. 7). Therefore, to study the *in vivo* behaviour of the BPQDs/PLGA

NSs, Cy5.5, a commonly used NIR fluorescent dye[61] was utilized to label the BPQDs/PLGA NSs by entrapping it into the NSs using the oil-in-water emulsion solvent evaporation method mentioned above. The synthesized Cy5.5-labelled BPQDs/PLGA NSs with similar size of the BPQDs/PLGA NSs exhibit bright and stable fluorescence at about 695 nm (Supplementary Fig. 8) enabling non-invasive monitoring and quantitative examination of the NSs biodistribution in the mice. Hence, the Balb/c nude mice bearing MCF7 breast tumours are intravenously injected with the Cy5.5-labelled BPQDs/PLGA NSs (100 μl of 1 mg BP ml$^{-1}$ for each mouse) for the biodistribution examinations.

The pharmacokinetics profile of the Cy5.5-labelled BPQDs/PLGA NSs was examined by fluorometry to determine the concentrations in blood at different time intervals post-injection (Fig. 6a). Blood circulation of the NSs obeys the typical two compartment model. After the first phase (distribution phase,

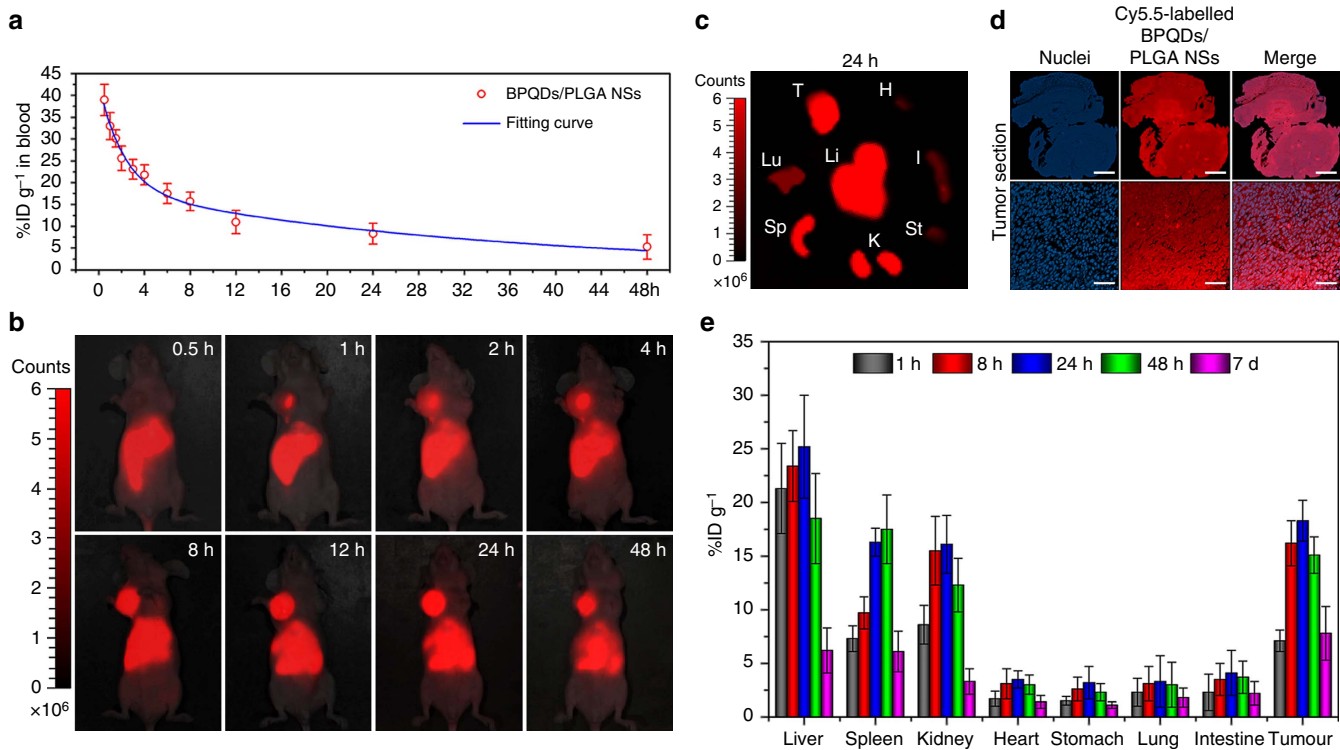

**Figure 6 | Pharmacokinetic and biodistribution analysis. (a)** Blood circulation curve of the Cy5.5-labelled BPQDs/PLGA NSs determined by measuring the Cy5.5 fluorescence intensity in the blood of the MCF7 tumour-bearing Balb/c mice at different time points post-injection of the NSs. The pharmacokinetics obeys a typical two compartment model (as shown by the fitted curve). **(b)** *In vivo* fluorescence images of the NSs-treated mice at different time points post-injection. **(c)** *Ex vivo* fluorescence images of the tumour and major organs from the NSs-treated mice at 24 h post-injection. H, heart; I, intestine; K, kidney; Lu, lung; Li, liver; Sp, spleen; St, stomach; T, tumour. **(d)** Fluorescence microscopy images of the tumour sections at macro-organizational level (scale bars, 1 mm) and micro-organizational level (scale bars, 50 μm) from the NSs-treated mice. The NSs are shown in red and the nuclei are shown in blue by staining with DAPI. **(e)** Quantitative biodistribution analysis of the NSs in mice by measuring the Cy5.5 fluorescence intensity in the tumours and major organs at different time points post-injection.

with a rapid decline) with a half-life of only $1.50 \pm 0.21$ h, the NSs in circulating blood exhibit a long second phase (elimination phase, the predominant process for NSs clearance) with a half-life of $22.66 \pm 3.65$ h. The volume of distribution ($V$) is measured to be $2.31 \pm 0.72$ ml and the area under curve (AUC) is $0.65 \pm 0.11$ mg h ml$^{-1}$. The long blood circulation of the NSs delays the macrophage clearance in reticuloendothelial systems[62], favouring enhanced tumour targeting by the EPR effect.

The biodistribution of the Cy5.5-labelled BPQDs/PLGA NSs in the mice is directly observed by fluorescence imaging. As shown in Fig. 6b, considerable fluorescence can be observed from the tumour at 1 h post-injection and the subcutaneous tumour can be definitely delineated from the other tissues. The fluorescence intensity in the tumour gradually increases up to 24 h, indicating that the NSs can continuously accumulate at the tumour site. At 48 h post-injection, the tumour still maintains strong fluorescence, suggesting good retention of the NSs in the tumour. Figure 6c shows the *ex vivo* fluorescence images obtained at 24 h post-injection, in which bright fluorescence can be observed from the tumour and some organs including the liver, spleen and kidney. Macro-organizational examination of a tumour ($\sim 50$ mm$^2$) in Fig. 6d (up) shows that the NSs are distributed throughout the entire tumour section. Moreover, Fig. 6d highlights the significant colocalization of the nuclei (DAPI staining, shown in blue) and NSs (shown in red) in the tumour section, confirming efficient penetration of the NSs within the tumour.

A quantitative biodistribution analysis of the Cy5.5-labelled BPQDs/PLGA NSs in mice is conducted (Fig. 6e). The tumour

and major organs were collected from the mice, weighed and solubilized by a lysis buffer at different time intervals post-injection. The homogenized tissue lysates were diluted and measured by fluorometry to quantitatively determine the NSs concentrations. At 24 h post-injection, large NSs concentrations can be found from not only the tumour, but also organs including the liver, spleen and kidney as consistent with the above *ex vivo* fluorescence examination. Uptake of the NSs by the liver and spleen may be due to reticuloendothelial system absorption[62], while the kidney uptake can be associated with possible renal excretion[63]. Even so, considerable uptake of the NSs by the tumour can be achieved on account of the EPR effect[21].

Since the NSs in the physiological medium can maintain their integrity (Fig. 3c,d) without causing evident fluorescence decrease of the entrapped Cy5.5 (Supplementary Fig. 8) for 7 days, the fluorescence examinations were further used to estimate the time-dependent residual amounts of the NSs in mice during the 7 days post-injection. The residual ratios were calculated by normalizing the total residual amounts in these organs and tissues to initial total amounts. It can be calculated that the residual ratio of the NSs decreases from 90.1% ID g$^{-1}$ at day 1 (24 h) to only 29.9% ID g$^{-1}$ at day 7, suggesting the possibility of the NSs to be partially metabolized. It is known that such nanoparticles is generally difficult to be completely metabolized and excreted from the body directly. However, the aforementioned biodegradability of the NSs enables harmless clearance from the body in a reasonable period of time (for example, several months).

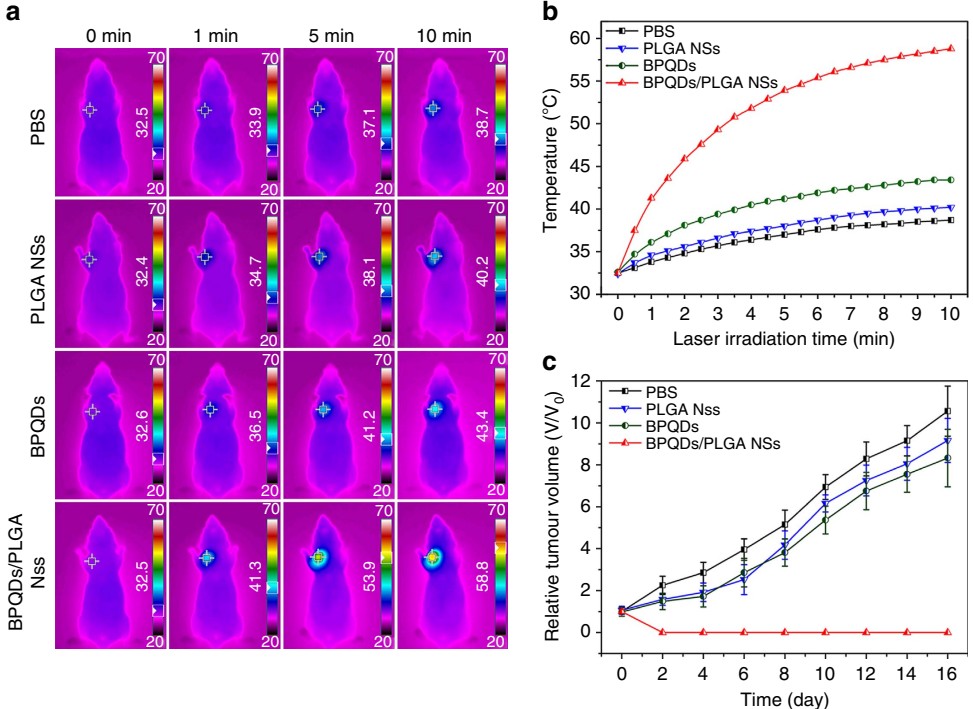

**Figure 7 | In vivo photothermal cancer therapy.** (**a**) Infrared thermographic maps and (**b**) Time-dependent temperature increase in the MCF7 breast tumour-bearing nude mice irradiated by the 808 nm laser ($1\,W\,cm^{-2}$) at 24 h after separate intravenous injection with 100 μl of PBS, PLGA NSs, BPQDs ($1\,mg\,ml^{-1}$) and BPQDs/PLGA NSs ($1\,mg\,BP\,ml^{-1}$) with the colour bar referring to the relative temperature. (**c**) Growth curves of MCF7 breast tumour in different groups of nude mice treated with PBS, PLGA NSs, BPQDs ($1\,mg\,ml^{-1}$) and BPQDs/PLGA NSs ($1\,mg\,BP\,ml^{-1}$) with the NIR laser irradiation.

**In vivo photothermal cancer therapy.** To evaluate the potential of the BPQDs/PLGA NSs in cancer PTT *in vivo*, MCF7 breast tumours were established *in situ* in the Balb/c nude mice. After the tumour volume has reached $\sim200\,mm^3$, 20 mice were randomly divided into four groups and aliquots (100 μl) of PBS, PLGA NSs, BPQDs ($1\,mg\,ml^{-1}$) and BPQDs/PLGA NSs ($1\,mg\,BP\,ml^{-1}$) were injected separately *via* the tail vein. At 24 h post-injection, the mice were anaesthetized and the entire region of the tumour was irradiated with the 808 nm laser ($1\,W\,cm^{-2}$) for 10 min. To monitor the photothermal effects *in vivo*, the infrared thermographic maps and changes in the tumour temperature were recorded by an infrared thermal imaging camera simultaneously (Fig. 7a,b). Under NIR laser irradiation, the tumour temperature of the mice injected with the bare BPQDs increases by only 10.8 °C which is just slightly higher than that of two other control groups of mice injected with the PLGA NSs (7.8 °C) and PBS (6.2 °C). In contrast, with regard to the mice injected with the BPQDs/PLGA NSs, the tumour temperature rises rapidly by 26.3 °C within 10 min of the NIR laser irradiation and the maximum temperature reaches 58.8 °C, which is high enough for tumour ablation[64]. The AuNRs were employed as a positive control in the photothermal experiments (Supplementary Fig. 9). Under the same irradiation condition, the tumour temperature of the mice injected with 100 μl of the AuNRs ($3.62\,mg\,ml^{-1}$) increases to 54.4 °C, which is lower than that induced by the NSs. These results indicate the high efficiency of the BPQDs/PLGA NSs as a PTT agent for *in vivo* tumour ablation.

To further evaluate the influence of PLGA encapsulation on the *in vivo* PTT efficiency of the BPQDs, the BPQDs and BPQDs/PLGA NSs with different concentrations (0.5, 1, 2 and 3 mg BP ml$^{-1}$) were injected intravenously into the tumour-bearing nude mice, which were irradiated with the NIR laser at different times (1, 4, 8, 12, 24 and 48 h) post-injection. As shown

in Supplementary Fig. 10, the BPQDs/PLGA NSs produce larger tumour temperature increase than the bare BPQDs under all conditions. The better *in vivo* PTT efficiency of the BPQDs/PLGA NSs than bare BPQDs pertaining to tumour ablation can be attributed to two factors. First, the BPQDs/PLGA NSs have better stability than the bare BPQDs and so can maintain the photothermal performance during circulation in the body. Second, compared with the ultrasmall BPQDs with a size of $\sim3\,nm$ (ref. 16), the size of the BPQDs/PLGA NSs of $\sim100\,nm$ is more appropriate for efficient tumour targeting and retension during the long blood circulation in the body.

After the above photothermal treatments with different samples, the tumour size was measured every 2 days after the treatment (Supplementary Table 1) and no obvious sign of toxic side effects such as abnormal body weight, activity, eating, drinking or neurological issues could be observed from all the experimental groups. The tumour volume variations determined from the different groups are consistent with those of temperature (Fig. 7c). The mice injected with the BPQDs/PLGA NSs and irradiated by the NIR laser, their tumours shrink gradually, are obliterated only leaving black scars, and then are completely cured in $\sim16$ days. More importantly, not even a single case of recurrence is observed. With regard to the mice treated with the bare BPQDs, the tumour growth is not inhibited effectively and so are the mice treated with the PLGA NSs or PBS alone. The mice treated with the BPQDs/PLGA NSs are tumour-free after the treatment and all survive for over 40 days until killed. In contrast, the mice in the other groups have a life span of 18–26 days and the tumour volume is as large as 1,600 mm$^3$. These results demonstrate the excellent efficacy of the NSs for photothermal cancer therapy of the BALB/c nude mice which have a deficient immune system. To further demonstrate the applicability of the NSs-mediated PTT, BALB/c mice, with a competent immune system, were employed as another animal

model in the photothermal treatments. As shown in Supplementary Fig. 11, the BPQDs/PLGA NSs also exhibit excellent photothermal efficacy to kill the melanoma tumour in the BALB/c mice without causing obvious toxic side effects. The results demonstrate that the NSs-mediated PTT is suitable for such two kinds of animals with different immune systems.

The anticancer efficiency is further analysed by a terminal deoxynucleotidyl transferase-mediated deoxyuridine triphosphate nick end-labelling (TUNEL) assay, which is generally utilized to detect the intratumoral levels of apoptosis. As shown in Supplementary Fig. 12, no or few TUNEL-positive cells (shown in green) are observed from the PBS, PLGA and BPQDs groups, while significant colocalization of nuclei (DAPI staining, shown in blue) and TUNEL-positive apoptotic cells (shown in green) can be observed from the BPQDs/PLGA NSs group. Moreover, the apoptosis examination at macro-organizational level ($\sim 10\,mm^2$) shows that the TUNEL-positive apoptotic cells are distributed throughout the tumour section (Supplementary Fig. 13). These results indicate that the NSs-mediated PTT can induce cancer cell death by activating apoptosis in the tumour.

The depth of photothermal damage is investigated in the tumour-bearing nude mice with the tumour volume as large as $1,000\,mm^3$. After the photothermal treatment, intratumoral apoptosis of the tumour sections at different depths was detected by a TUNEL assay (Supplementary Fig. 14). Most cancer cells undergo apoptosis in the tumour sections at depths of no more than 6 mm. Although evident depth-dependent decay of the PTT efficiency is observed when the depth is over 6 mm, significant apoptosis of cells can still be found from the section at the depth of 10 mm. The considerable photothermal damage to deep tissues stems from the excellent PTT efficiency of the NSs and high tissue penetration ability of NIR light. It should be noted that although the penetration depth of NIR light is limited to be no more than 10 mm (ref. 21), clinical photothermal treatment of deep tumours is still achievable with the aid of specialized medical devices such as endoscopic ones in combination with optical fibres as well as implanted NIR devices[65,66].

## Discussion

BPQDs/PLGA NSs with highly efficient photothermal performance are fabricated using PLGA loaded with 3 nm BPQDs by an oil-in-water emulsion solvent evaporation method. Owing to the strong hydrophobicity of PLGA, rapid degradation of the BPQDs is prevented so that the photothermal performance of the BPQDs/PLGA NSs in the physiological environment is improved significantly. No appreciable toxicity and good biocompatibility are observed from the BPQDs/PLGA NSs based on the haematological, blood biochemical and histological analyses. Boasting long blood circulation in the body, the NSs after intravenous injection show efficient tumour accumulation by the EPR effect. By activating apoptosis of cancer cells even in the deep tumour tissues, the NSs give rise to highly efficient tumour ablation under NIR laser irradiation, and the NSs-mediated PTT is proven to be suitable for two kinds of animals with different immune systems.

Compared with previously reported nanoagents, these BPQDs/ PLGA NSs are especially attractive due to the unique biodegradability (as illustrated in Supplementary Fig. 15). PLGA, which is FDA-approved degrades by hydrolysis forming carbon dioxide and water eventually in a reasonable time (for example, several months). After intravenous administration into the body, the BPQDs/PLGA NSs with size of $\sim 100$ nm can circumvent rapid degradation of the BPQDs in the optical performance during long-time circulation (for example, 24 h), and ensure sufficient tumour accumulation for highly efficient photothermal cancer

therapy. After fulfilling their therapeutic functions, degradation of PLGA disrupts the NSs causing the release of the BPQDs. The BPQDs then degrade to form nontoxic phosphate and phosphonate enabling harmless clearance from the body. In addition, such slow degradation of the BPQDs circumvents the local imbalance of phosphorus in vivo. Therefore, such BP-based PTT agent with the unique combination of biodegradability and biocompatibility has immense clinical potential.

## Methods

**Materials.** The BP crystals were purchased from a commercial supplier (Smart-Elements) and stored in a dark Ar glovebox and N-methyl-2-pyrrolidone (NMP) (99.5%, anhydrous) was obtained from Aladdin Reagents. The PLGA (50:50, MW: 40,000–70,000), polyvinyl alcohol (MW: 9,000–10,000), and dichloromethane (DCM) were purchased from Sigma-Aldrich (Santa Barbara, USA). The PBS (pH 7.4), foetal bovine serum, H-DMEM, trypsin-EDTA, Calcein AM and PI were obtained from Gibco Life Technologies (AG, Switzerland). The Cy5.5 NIR fluorescence dye was purchased from Lumiprobe (US). All the chemicals used in this study were analytical reagent grade and used without further purification.

**Synthesis of BPQDs.** The BPQDs were prepared by a simple liquid exfoliation technique described in the literature[39]. In brief, 20 mg of the bulk BP powders were dispersed in 20 ml of NMP and sonicated with a sonic tip (ultrasonic frequency: 19–25 kHz) for 4 h (period of 2 s with the interval of 4 s) using a power of 1,200 W. The mixture was sonicated overnight in an ice bath using a power of 300 W. The dispersion was centrifuged for 20 min at 7,000 r.p.m. and the supernatant containing the BPQDs was decanted gently.

**Preparation of BPQDs/PLGA NSs.** The BPQDs/PLGA NSs were prepared by an oil-in-water emulsion solvent evaporation method. IN brief, 10 ml of the BPQDs suspension in NMP (200 p.p.m.) were centrifuged for 20 min at 12,000 r.p.m. and the precipitate was redispersed in 1 ml of the PLGA solution in DCM with a concentration of $10\,mg\,ml^{-1}$. After probe sonication for 5 min, the mixture was dispersed in 10 ml of 0.5% (w/v) polyvinyl alcohol aqueous solution and sonicated for another 5 min. The emulsion was stirred overnight at room temperature to evaporate the residual DCM. Finally, the NS suspension was centrifuged at 7,000 r.p.m. for 15 min, washed twice with deionized water to remove dissociated BPQDs, and re-suspended in the aqueous solution.

**Morphology and characterization.** The TEM images were acquired on the JEOL JEM-2010 TEM at an acceleration voltage of 200 kV and AFM was performed on an MFP-3D-S AFM (Asylum Research, USA) using the tapping mode in air (NanoSensors SSS-NCH probe with the tip radius as small as 2 nm). The BP concentration was determined by inductively coupled plasma atomic emission spectroscopy (IRIS Intrepid II XSP, thermo Electron Corporation, USA). The SEM images were obtained on the field-emission SEM (NOVA NANOSEM430, FEI, Netherlands) at 5–10 kV after gold coating for 120s (EM-SCD500, Leica, Germany). The energy dispersive X-ray spectroscopy was conducted on the Oxford INCA 300 equipped on the SEM. The Raman scattering was performed on the Horiba Jobin-Yvon Lab Ram HR VIS high-resolution confocal Raman microscope with the 633 nm laser as the excitation source. The ultraviolet–visible-NIR absorption spectra were acquired on a TU-1810 ultraviolet–visible spectrophotometer (Purkinje General Instrument Co. Ltd. Beijing, China) using QS-grade quartz cuvettes at room temperature.

**NIR laser-induced heat conversion.** A fiber-coupled continuous semiconductor diode laser (808 nm, KS-810F-8000, Kai Site Electronic Technology Co., Ltd. Shaanxi, China) was employed in the experiments. Sample (1 ml) in a 1-cm path length quartz cuvette was irradiated by the laser with a power density of $1\,W\,cm^{-2}$ for 10 min. The laser spot was adjusted to cover the entire surface of the sample. Real-time thermal imaging was performed and the maximum temperature was recorded by the Fluke Ti27 infrared thermal imaging camera (USA).

***In vitro* degradation.** The BPQDs and the BPQDs/PLGA NSs with the same amount of BPQDs (10 p.p.m.) were dispersed in PBS (pH 7.4), kept in closed sample vials and maintained in a horizontal shaker at 37 °C. At predetermined time intervals, the BPQDs/PLGA NSs were centrifuged at 7,000 r.p.m. and dried under vacuum. The residual weight percentages of the sample for different degradation time (0–8 weeks) were determined by the following formula: residual weight (%) $= Wr \times 100/Wo$, where $Wr$ is the dry weight of the sample after degradation and $Wo$ is the initial weight of the sample. Degradation of the BPQDs/PLGA NSs after 808 nm laser ($1\,W\,cm^{-2}$) illumination for 10 min was also evaluated by the above method.

***In vitro* cytotoxicity assays.** The human skin fibroblast, MCF7 human breast cancer cells and B16 melanoma cells were purchased from China Type Culture Collection (CTCC) obtained from the American Type Culture Collection (ATCC). Cells were routinely tested for mycoplasma contamination using MycoSET Mycoplasma real-time PCR detection Kit (Life Technologies, Foster City, CA, USA). These cells were seeded on a 96-well plate ($1 \times 10^4$ cells per well) with 200 μl of H-DMEM (Gibco BRL) supplemented with 10% (v/v) foetal bovine serum and maintained in an incubator at 37 °C in a humidified atmosphere consisting of 5% $CO_2$. After culturing for 12 h, the cell culture medium was replaced with 200 μl of the H-DMEM medium containing different concentrations of BPQDs/PLGA NSs (internal BPQDs concentrations of 0, 2, 5, 10, 20, 50 and 100 p.p.m., respectively). Five multiple holes were set for every sample. The cell viability was quantitatively determined using the CCK-8 assay. The absorbance at 450 nm was determined on a microplate spectrophotometer (Varioskan Flash 4.00.53, Finland) as an indicator of viable cells. The cell viability was normalized to the control group without any treatment and the following formula was used to calculate cell growth inhibition: cell viability (%) = (mean of Abs. value of treatment group/mean Abs. value of control) × 100%.

***In vitro* photothermal experiments.** The MCF7 and B16 cells ($1 \times 10^4$ cells per well) were seeded on 96-well plates and incubated overnight in an incubator at 37 °C. After rinsing with PBS, the MCF7 and B16 cells were incubated with BPQDs/PLGA NSs (internal BPQDs concentrations of 0, 2, 5, 10 and 20 p.p.m.) for 4 h at 37 °C and then illuminated with the 808 nm laser (1 W cm$^{-2}$) for 10 min. The laser spot was adjusted to fully cover the area of each well. After irradiation, the cells were incubation for 12 h, rinsed with PBS, and co-stained with Calcein AM and PI for 30 min. Afterwards, the cells of the experimental group were rinsed with PBS and examined by an Olympus IX71 inverted fluorescence microscope. The cell viability was determined quantitatively by the CCK-8 assay and compared with the control group which did not undergo any treatment.

The AuNRs were employed as a positive control in the *in vitro* photothermal experiments. The MCF7 cells was incubated with AuNRs (concentrations of 0, 7.2, 18.1, 36.2 and 72.4 p.p.m.) for 4 h at 37 °C and then irradiated with the 808 nm laser (1 W cm$^{-2}$) for 10 min. The corresponding fluorescence images of the cells and cell viability were assessed by the above method.

***In vivo* toxicity.** Sixty healthy female Balb/c mice (6 weeks old) were obtained from Slac Laboratory Animal Co. Ltd (Hunan, China) and all the *in vivo* experiments followed the protocols approved by the Animal Care and Use Committee of the Shenzhen Institutes of Advanced Technology, Chinese Academy of Sciences. Aliquots (200 μl) of the BPQDs/PLGA NSs (1 mg BP ml$^{-1}$) were injected separately into the mice via the tail vein. The mice were randomly divided into 4 groups and subjected to variable conditions. This include: (1) control group without any treatment, (2) NSs directly intravenously injected into the mice, (3) NSs intravenously injected into the mice after 808 nm laser irradiation for 10 min and (4) NSs intravenously injected into the mice which are then exposed to artificial daylight for 24 h. The mice were killed at various time points after injection (1, 7 and 28 days, five mice per group at each time point). About 0.8 ml of blood were collected from each mouse to conduct complete blood panel analysis and serum biochemistry assay at the Shanghai Research Center for Biomodel Organism. The major organs (liver, spleen, kidney, heart and lung) were harvested, fixed in 10% neutral buffered formalin, processed routinely into paraffin, sectioned at 8 μm, stained with haematoxylin and eosin, and examined by digital microscopy.

**Pharmacokinetic and biodistribution analysis.** In the biodistribution and pharmacokinetic analysis, fluorescent labelled BPQDs/PLGA NSs were prepared by adding 0.1 mg ml$^{-1}$ of Cy5.5 to the BPQDs/PLGA solution in DCM followed by oil-in-water emulsion solvent evaporation described above. The excess dye molecules were removed by centrifugation and washed away with water more than 5 times until no noticeable colour change was observed from the supernatant fluids followed by resuspension in PBS.

The female Balb/c nude mice (6 weeks old) were purchased from Slac Laboratory Animal Co.Ltd (Hunan, China). In the pharmacokinetic analysis, blood circulation was assessed by drawing 10 μl of blood from the tail veins of the Balb/c nude mice at certain time intervals post-injection of the Cy5.5-labelled BPQDs/PLGA NSs. Each blood sample was dissolved in 1 ml of the lysis buffer (the same as the above used) and the concentration of the NSs in the blood was determined from the fluorescence spectrum acquired on a Fluoromax 4 fluorometer (Horiba Jobin Yvon, France). A series of dilutions were performed to obtain a standard calibration curve. The blank blood sample without injection was measured to determine the blood auto-fluorescence level, which was subtracted from the fluorescence intensities of injected samples during concentration calculation. The pharmacokinetic parameters such as half-life ($t_{1/2}$), V and AUC were determined using a Microsoft add-in tool, PKSolver[67].

In the *in vivo* fluorescence imaging experiments, the Balb/c nude mice bearing MCF7 breast tumours were intravenously injected with the Cy5.5-labelled BPQDs/PLGA NSs (100 μl of 1 mg BP ml$^{-1}$ for each mouse) and examined by a fluorescence (Xenogen IVIS-Spectrum) imaging system as a function of time for up to 48 h. NIR light with a peak wavelength of 675 nm was used as the excitation

source and *in vivo* spectral imaging with the Cy5.5 bandpass emission filter (680–720 nm) was carried out for an exposure time of 200 ms for each image frame. All the images were captured using identical system settings and auto-fluorescence was removed using the spectral unmixing software.

In the *ex vivo* fluorescence imaging experiments, the NSs-treated mice were killed by cervical dislocation and the corresponding major organs and tissues including the liver, spleen, kidney, heart, stomach, lung, intestine and tumour were collected and imaged immediately afterwards. The tumours were fixed in 10% neutral buffered formalin and embedded in paraffin. Sections of whole tumour were stained using DAPI (shown in blue) to label all nuclei of tumour cells. The fluorescence images of the tumour sections were acquired on the Leica DM4000B fluorescence microscope (Leica, Nussloch, Germany).

In the quantitative biodistribution analysis, the NSs-treated mice were killed and the organs/tissues were weighed and solubilized by a lysis buffer (1% SDS, 1% Triton X-100, 40 mM Tris Acetate) using a PowerGen homogenizer (Fisher Scientific). The clear homogeneous tissue lysates were diluted 100 times to avoid light scattering and self-quenching during fluorescence measurement. The fluorescence intensities of both the standard samples and real tissues were adjusted to be in the linear range by appropriate dilution and subjected to fluorometry to quantitatively determine the NSs concentrations. The organs and tissues from a control mouse without injection of the NSs were collected and used as controls to subtract the auto-fluorescence background in various tissues. The samples were measured in triplicate to ensure reproducibility and accuracy. The biodistribution of the NSs in the various organs of the mice was calculated and presented as the percentage of injected dose per gram of tissue (%ID g$^{-1}$).

***In vivo* photothermal experiments.** To establish the MCF7 breast tumours *in situ* in the Balb/c nude mice, $1 \times 10^7$ MCF7 cells in 100 μl PBS were subcutaneously injected into the left foreleg armpit of each mouse. When the tumour volume reached 200 mm$^3$, the mice were randomly divided into 4 groups ($n = 5$ per group) and aliquots (100 μl) of PBS, PLGA NSs, BPQDs (1 mg ml$^{-1}$) and BPQDs/PLGA NSs (1 mg BP ml$^{-1}$) were injected separately into the nude mice via the tail vein. At 24 h post-injection, the mice were anaesthetized and the entire region of the tumour was irradiated with the 808 nm NIR laser (1 W cm$^{-2}$) for 10 min. The temperature of the tumours and infrared thermographic maps were monitored by an infrared thermal imaging camera (Ti27, Fluke, USA) simultaneously. The AuNRs (100 μl, 3.62 mg ml$^{-1}$) were employed as a positive control in the photothermal experiments. After laser irradiation, the tumour size was measured by a caliper every other day according to the formula: volume ($V$) = (tumour length) × (tumour width)$^2$/2, and no mice died during the course of therapy. The same observer performed all tumour measurements in this study. The relative tumour volume was calculated as $V/V_0$ with $V_0$ being the initial tumour volume at the start of the treatment. Daily clinical observations including weekends and holidays were performed to monitor the animals for signs of distress. When the tumour size reached 20 mm in any direction or the mice displayed restriction, inability to access food and water, pressure on internal organs or sensitive regions of the body, the mice were euthanized. To further compare the photothermal effects between the BPQDs and BPQDs/PLGA NSs in details, photothermal experiments with different injection concentrations (0.5, 1, 2 and 3 mg BP ml$^{-1}$) and irradiation time (1, 4, 8, 12, 24 and 48 h post-injection) were performed. The temperature of the tumours and infrared thermographic maps were obtained by the infrared thermal imaging camera.

Another tumour model with competent immune system was established *in situ* in the Balb/c mice by subcutaneously injection of $1 \times 10^7$ B16 melanoma cells in PBS to the left rear flank of each mouse. When the tumour volume reached 100 mm$^3$, aliquots (100 μl) of PBS, PLGA NSs, BPQDs (1 mg ml$^{-1}$), and BPQDs/PLGA NSs (1 mg BP ml$^{-1}$) were injected separately into the mice via the tail vein. At 24 h post-injection, *in vivo* photothermal experiments were conducted as described above.

**Apoptosis detection.** The tumours were collected from the Balb/c nude mice treated with PBS, PLGA NSs, BPQDs and BPQDs/PLGA NSs 24 h after the treatment. The individual tumours were fixed in 10% neutral buffered formalin, embedded in paraffin, sectioned at 5 micrometres and stained using the TUNEL technique using the *In Situ* Cell Death Detection Kit (Roche Applied Science, Germany). The experimental procedures were in accordance with the manufacturer's instructions. DAPI was used to stain the sections in the absence of light to label the apoptotic cells and cellular DNA. The fluorescence images were taken on the Leica DM4000B fluorescence microscope (Leica, Nussloch, Germany). To further investigate the depth of the photothermal damage, the tumour-bearing nude mice with a tumour volume as large as 1,000 mm$^3$ was illuminated with 808 nm NIR laser (1 W cm$^{-2}$) for 10 min. Intratumoral apoptosis of the tumour sections was monitored at different depths (2, 4, 6, 8 and 10 mm) by the TUNEL assay.

**Statistical analysis.** All the data were presented as means ± s.d. To test the significance of the observed differences between the study groups, analysis by variance statistics was applied and a value of $P < 0.05$ was considered to be statistically significant.

**Data availability**. The data that support the findings of this study are available from the corresponding author upon reasonable request.

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

## Acknowledgements

We acknowledge financial support from the National Natural Science Foundation of China (51372175, 61435010), Leading Talents of Guangdong province Program (00201520), Science and Technology Innovation Commission of Shenzhen (KQTD2015032416270385, JCYJ20160229195124187), as well as Hong Kong Research Grants Council (RGC) General Research Funds (GRF) No. CityU 11301215.

## Author contributions

X.F.Y., H.Z. and P.K.C. conceived the idea and designed the experiments. J.S. and H.X. performed mainly the syntheses, experimental measurements and data analysis. H.H., Z.L., Z.S., Y.X., Q.X. and Y.Z. assisted with data analysis and interpretation. J.S. drafted the paper and X.F.Y., H.W., H.Z. and P.K.C. provided revisions. All authors approved the final version of the paper for submission.

## Additional information

**Competing financial interests:** The authors declare no competing financial interests.

