## [Peer Review File · Nature Communications]

Reviewer #1 (Remarks to the Author):

The authors present an interesting method for the fabrication of a biodegradable photothermal therapy agent based on black phosphorus. As a new member of 2D materials, black phosphorus has already attracted tremendous interest within the past year. This work is a significant contribution to black phosphorus for their new applications in the field of biomedicine. As a photothermal therapy agent, these BP vesicles are especially attractive due to their unique biodegradability and excellent biocompatibility, while clinical adoption of various other nanoagents have been stifled by unsolved concerns such as biodegradability as well as long-term toxicity. On the other hand, since the lack of air- and water-stability is a fundamental obstacle hampering application of black phosphorus, this work provides an efficient strategy to enhance the stability of it. Overall, this work is well presented and the results are appealing. I could recommend publication of this paper after addressing the following issues.

1. BP normally has a resonant absorption at longer wavelength (1300 nm-1600 nm). However, in this work, the laser with a wavelength of 808 nm was used for the photothermal treatment. The samples have pretty low absorption at this wavelength. Please explain why not use lasers which match the resonant absorption.
2. It is difficult for readers to see clearly of the nanospheres in Figure 1f, a higher quality SEM image would be appreciated.
3. Figure 1d, the height number for vertical scale is missing.
4. The authors need to address why BPQDs/PLGA NSs (Figure 2c) shows much stronger absorption than BPQDs (Figure 2b) at appropriate position.
5. In Figure 3c, what does the asterisk (*) mean? Authors should explain it in the figure legend or figure caption.
6. The authors have mentioned that the degradation rate of the PLGA and BPQDs can be controlled in the "Introduction" section. Please elaborate how to control their degradation rate.

Reviewer #2 (Remarks to the Author):

Photothermal therapies have demonstrated high potential for selective treatment of malignancies with reduced side effects. In "Biodegradable black phosphorus vesicles for in vivo photothermal cancer therapy", Dr. Xue-Feng Yu, Dr. Han Zhang and Dr. Paul Chu, and their collaborators prepared polymeric nanoparticles based on poly(lactic-co-glycolic acid) (PLGA) for delivering black phosphorous quantum dots (BPQDs) directed to enhanced antitumor activity. The novelty of the work is arguable as drug delivery systems from PLGA loading and protecting a broad range of bioactive molecules have been widely reported, while several works, including one paper from the authors, have already considered the use of black phosphorous for specific tumor therapy (H. Wang, et al. *J. Am. Chem. Soc.* 137 (2015) 11376-11382; N. M. Latiff, et al. *Chemistry* 21 (2015) 13991-13995; X. Zhang, et al. *Angew Chemie*, 127 (2015) 11688-11692; H.U. Lee, et al. *Small* 12 (2016) 214-219). Moreover, the demonstration of the therapeutic benefit for the developed polymeric nanoparticles is biased, as the dosing and timing for irradiation are more suitable for the nanoparticle formulation than for the BPQDs. Therefore, this reviewer suggests the following revisions to improve their significance of this work:

1. The use of the term "vesicle" for describing these particles should be reconsidered. Vesicles present inner compartments including solvent. The BPQDs-loaded nanoparticles have solid core. The term "Nanospheres" is preferable.
2. The in vitro and in vivo activity studies should include a positive control, such as gold nanorods.
3. The authors should determine the pharmacokinetics and tissue distribution of the nanoparticles, as well as BPQDs.
4. The intratumoral microdistribution of the particles and the intratumoral levels of apoptosis should be addressed, as it could be that the photothermal damage is solely occurring on the blood

vessels if the particles do not penetrate within the tumor, resulting in a vascular disruption therapy rather than photothermal ablation.

5. The *in vivo* study was done against MCF7 tumors for 14 days and the mice showed tumor free survival for 40 days. However, the pictures in figure 5c do not support these observations as the mice treated with the particles still show large tumors. Moreover, MCF7 tumor model may not be appropriate for studying photothermal therapy, as it is prepared in mice with a deficient immune system. Thus, the activity of the particles should be studied in a second model with competent immune system and relevant to photothermal therapy to validate the applicability of the particles.

6. Photothermal therapy is limited by the penetration of light in tissues. Therefore, it is mainly applied to superficial tumors. Improving the depth for photothermal damage can result in innovative cancer treatments. Therefore, I recommend the authors to check if these particles can improve the damage in deep tissues.

7. The toxicity from the particles was completely ignored. The authors should demonstrate the safety of these particles without irradiation, with irradiation, and by exposing the mice to daylight (as a main issue for photosensitizer molecules is the side effects arising after sun exposure).

8. The authors mentioned that the higher efficacy of their particles over the BPQDs is because the 100 nm size of the nanoparticles is more appropriate for efficient tumor targeting compared to the ultrasmall size of BPQDs (3 nm). However, this observation is biased from the experimental setting. Accordingly, the dose and time for irradiation (24 h after injection) of the antitumor experiment were more suitable for the nanoparticles than for the BPQDs. If the experiment is performed with higher BPQDs doses and earlier irradiations, the results could provide better antitumor effect for BPQDs than for PLGA/BPQDs, making the development of PLGA/BPQDs unnecessary. The authors should demonstrate that at the optimal conditions for each formulation, i.e. dose and irradiation timing for BPQDs and PLGA/BPQDs, the PLGA/BPQDs still perform better than BPQDs.

Minor points:

1. English should be carefully revised. There are too many misspellings throughout the manuscript.

Reviewer #3 (Remarks to the Author):

A. Summary of the key results

The authors constructed a biodegradable black phosphorus (BP) vesicle using a widely used vehicle, poly (lactic-co-glycolic acid) (PLGA), loaded with BP quantum dots (BPQDs). This new BPQDs/PLGA nanosphere showed good photothermal efficiency and excellent tumor targeting ability. Although black phosphorus quantum dots (BPQDs) has been showed a potential application in clinical treatment, due to its unique biodegradability and excellent biocompatibility.

B. Originality and interest: if not novel, please give references

Here, the biodegradable black phosphorus (BP) quantum dots (BPQDs) is novel for biomedical application. However, the unique biodegradability and excellent biocompatibility *in vivo* is not clear due to the limited data.

C. Data & methodology: validity of approach, quality of data, quality of presentation

The validity of approach is high; the quality of data is high; the quality of presentation is high.

D. Appropriate use of statistics and treatment of uncertainties

Not mentioned in this paper.

E. Conclusions: robustness, validity, reliability

The data can not support the conclusions very well, especially the excellent tumor targeting ability and biocompatibility of BPQDs/PLGA nanosphere *in vivo*. Suggested improvements: experiments, data for possible revision

F. References: appropriate credit to previous work?

Good.

G. Clarity and context: lucidity of abstract/summary, appropriateness of abstract, introduction and

conclusions
Good.

Personal comments:

1. The data can not support the conclusions very well, especially the excellent tumor targeting ability and biocompatibility of BPQDs/PLGA nanosphere in vivo. In addition, the in vivo studies on the unique biodegradability and excellent biocompatibility of BPQDs/PLGA nanosphere are important for its further application. Therefore, the distribution and metabolism of BPQDs/PLGA nanosphere in mice should be investigated.
2. Although BPQDs have a broad absorption across the ultraviolet (UV) and infrared regions, the common PPT agents, such as gold nanorods and carbon nanotube, may have higher photothermal conversion efficiency in NIR, because of their specific absorption peak.
3. The concentrations of BPQDs/PLGA nanosphere and the laser doses in Fig. 2-3 are not clear.
4. The BPQDs/PLGA nanosphere may mainly accumulate in liver after i.v. injection into mice, therefore, the toxic analysis in liver and other organs should be determined.
5. BPQDs/PLGA nanosphere were used as photothermal agent, therefore, the stability or biodegradation after the laser treatment should be shown to further understand the metabolism of BPQDs/PLGA nanosphere in vivo.
6. The laser treatment should be labeled in the figures 4-5.
7. The author should describe the statistics method in the Methods Section.

Revisions and list of changes (NCOMMS-16-03858)

Replies to the 1st reviewer's comments (NCOMMS-16-03858)

Comment 1: BP normally has a resonant absorption at longer wavelength (1300 nm-1600 nm). However, in this work, the laser with a wavelength of 808 nm was used for the photothermal treatment. The samples have pretty low absorption at this wavelength. Please explain why not use lasers which match the resonant absorption.

Reply 1: We agree that 808 nm is not the wavelength corresponding to the strongest absorption of BP. However, the 808 nm light is mostly used in previously reported photothermal experiments and the 808 nm semiconductor laser is common and cheap. Here, to facilitate direct comparison of the photothermal performance of our samples with previously reported photothermal agents, we have chosen the 808 nm light in our photothermal experiments.

Comment 2: It is difficult for readers to see clearly of the nanospheres in Figure 1f, a higher quality SEM image would be appreciated.

Reply 2: A high-quality SEM image has been substituted for the old one in Figure 1f.

Changed in the revised manuscript (Figure 1f):

Comment 3: Figure 1d, the height number for vertical scale is missing.

Reply 3: The height number for the vertical scale in Figure 1d has been added.

Changed in the revised manuscript (Figure 1d):

Comment 4: The authors need to address why BPQDs/PLGA NSs (Figure 2c) shows much stronger absorption than BPQDs (Figure 2b) at appropriate position.

Reply 4: The BPQDs/PLGA NSs show stronger absorption than the BPQDs due to the contribution of the PLGA shells and the larger particle. We have added the corresponding description in the revised manuscript.

Added in the revised manuscript (Line 4, Page 7):

“In contrast, the BPQDs/PLGA NSs show stronger absorption than the BPQDs due to the contribution of the PLGA shells and the larger size of the whole particle.”

Comment 5: In Figure 3c, what does the asterisk (*) mean? Authors should explain it in the figure legend or figure caption.

Reply 4: The asterisk (*) means the *P* value obtained from the statistical analysis performed by the analysis of variance (ANOVA). We have explained it in the figure legend of Figure 3 and added the corresponding description in the revised manuscript.

Added in the figure legend of Figure 3:

“(n=5; * $P < 0.05$, ** $P < 0.01$, *** $P < 0.001$; ANOVA)”

Added in the “Methods” section (Line 12, Page 25):

“**Statistical analysis.** All the data were presented as means \pm standard deviation (SD). In order to test the significance of the observed differences between the study groups, analysis by variance (ANOVA) statistics was applied and a value of $P < 0.05$ was considered to be statistically significant.”

Comment 6: The authors have mentioned that the degradation rate of the PLGA and BPQDs can be controlled in the "Introduction" section. Please elaborate how to control their degradation rate.

Reply 6: Thank you for your suggestion. We have added the description about how to control the degradation rate of the PLGA in the revised manuscript.

Added in the revised manuscript (Line 19, Page 4):

“In general, the degradation period of PLGA spans several months and its degradation rate can be controlled by adjusting the chemical composition such as the monomer ratio and molecular weight⁵².”

Replies to the 2nd reviewer's comments (NCOMMS-16-03858)

Comment 1: The use of the term "vesicle" for describing these particles should be reconsidered. Vesicles present inner compartments including solvent. The BPQDs-loaded nanoparticles have solid core. The term "Nanospheres" is preferable.

Reply 1: Thank you for your suggestion. The term "vesicles" has been changed to “nanospheres” in the revised manuscript.

Changed in the revised manuscript:

“vesicles” has been changed to “nanospheres”

Comment 2: The *in vitro* and *in vivo* activity studies should include a positive control, such as gold nanorods.

Reply 2: Per your suggestion, we have added the *in vitro* and *in vivo* activity of the AuNRs as a positive control. The *in vitro* and *in vivo* photothermal performance of the BPQDs/PLGA nanospheres is compared with that of AuNRs. The corresponding results and discussions have been added in the revised manuscript.

Added in the Supplementary Information (Supplementary Figure 6):

Supplementary Figure 6 | Comparison of photothermal performance between the BPQDs/PLGA NSs and AuNRs. (a) Absorption spectra and (b) Photothermal heating curves of the BPQDs/PLGA NSs (20.0 ppm BPQDs) and AuNRs (72.4 ppm) with the same absorption at 808 nm. The AuNRs as the positive control are synthesized in an aqueous solution by a common seed-mediated growth method reported previously^{1,2} and BSA is used to conjugate with the AuNRs using a method described previously^{2,3}. (c) Relative viability of the MCF7 cells after incubation with BPQDs/PLGA NSs and AuNRs with different concentrations (same absorption at 808 nm) for 4 h and irradiated with the 808 nm laser (1 W/cm²) for 10 min. (d) Corresponding fluorescence images of the cells stained with calcein AM (live cells, green fluorescence) and PI (dead cells, red fluorescence).

Added in the revised manuscript (Line 2, Page 10):

“In the next step, the PTT efficiency of the BPQDs/PLGA NSs was compared with that of gold nanorods (AuNRs), one of the common photothermal agents. On account of the large NIR extinction coefficient and high photothermal conversion efficiency of the BPQDs³⁹, the BPQDs/PLGA NSs are more efficient in increasing the solution temperature than the AuNRs (Supplementary Fig. 6). In the cell photothermal experiments, both of the NSs (containing only 10 ppm of BPQDs) and AuNRs (72.4 ppm) can kill the cancer cells almost completely, but it is clear that less BPQDs are needed. These results confirm the suitability of BPQDs/PLGA NSs as an efficient PTT agent.”

Added in the Supplementary Information (Supplementary Figure 9):

Supplementary Figure 9 | Photothermal performance comparison of the BPQDs/PLGA NSs and AuNRs for *in vivo* tumor ablation. Infrared thermographic maps of the tumor-bearing nude mice irradiated by the 808 nm laser (1 W/cm²) at 24 h after separate intravenous injection with 100 μL of

BPQDs/PLGA NSs (1 mg BP/mL) and AuNRs (3.62 mg/mL) with the color bar referring to the relative temperature.

Added in the revised manuscript (Line 13, Page 14):

“The AuNRs were employed as a positive control in the photothermal experiments (Supplementary Fig. 9). Under the same irradiation condition, the tumor temperature of the mice injected with 100 μ L of the AuNRs (3.62 mg/mL) increases to 54.4 $^{\circ}$ C, which is lower than that induced by the NSs. These results indicate the high efficiency of the BPQDs/PLGA NSs as a PTT agent for *in vivo* tumor ablation.”

Added in the “Methods” section (Line 7, Page 21):

“The AuNRs were employed as a positive control in the *in vitro* photothermal experiments. The MCF7 cells was incubated with AuNRs (concentrations of 0, 7.2, 18.1, 36.2, and 72.4 ppm) for 4 h at 37 $^{\circ}$ C and then irradiated with the 808 nm laser (1 W/cm²) for 10 min. The corresponding fluorescence images of the cells and cell viability were assessed by the above method.”

Added in the “Methods” section (Line 8, Page 24):

“The AuNRs (100 μ L, 3.62 mg/mL) were employed as a positive control in the photothermal experiments.”

Added in the Supplementary Information (Supplementary References 1-3, Page 16):

1. Shao, J. *et al.* PLLA nanofibrous paper-based plasmonic substrate with tailored hydrophilicity for focusing SERS detection. *ACS Appl. Mater. Interfaces* **7**, 5391-5399 (2015).
2. Li, Z. *et al.* Small gold nanorods laden macrophages for enhanced tumor coverage in photothermal therapy. *Biomaterials* **74**, 144-154 (2016).
3. Ali, M. R. K., Panikkanvalappil, S. R., & El-Sayed, M. A. Enhancing the efficiency of gold nanoparticles treatment of cancer by increasing their rate of endocytosis and cell accumulation using rifampicin. *J. Am. Chem. Soc.* **136**, 4464-4467 (2014).

Comment 3: The authors should determine the pharmacokinetics and tissue distribution of the nanoparticles, as well as BPQDs.

Reply 3: Per your suggestion, we have performed additional experiments to examine the pharmacokinetics and tissue distribution of the BPQDs/PLGA nanospheres by labeling them with a

commonly used NIR fluorescent dye. For the BPQDs, we have also tried to examine their distribution by using ICP-AES. The corresponding results and discussion have been added in the revised manuscript.

Added in the Supplementary Information (Supplementary Figure 7):

Supplementary Figure 7 | *In vivo* biodistribution analysis of BPQDs. The concentrations of P in tumor and major organs of MCF7 breast tumor-bearing mice without or with BPQDs injection (5 mg/kg) are determined by ICP-AES. Since the original P concentrations in the organs are much larger than the injection dose, no meaningful difference can be observed from the P concentrations in all the organs after injection of the BPQDs.

Added in the Supplementary Information (Supplementary Figure 8):

Supplementary Figure 8 | Characterization of the Cy5.5-labeled BPQDs/PLGA NSs. (a) SEM image of the Cy5.5-labeled BPQDs/PLGA NSs. (b) Photographs of the BPQDs/PLGA NSs and Cy5.5-labeled BPQDs/PLGA NSs. (c) Emission spectra ($\lambda_{ex} = 675$ nm) of the Cy5.5-labeled BPQDs/PLGA NSs before and after storage in PBS for 7 days. No fluorescence can be observed from the BPQDs/PLGA NSs without Cy5.5 labeling.

Added in the revised manuscript (Figure 6):

Figure 6 | Pharmacokinetic and biodistribution analysis. (a) Blood circulation curve of the Cy5.5-labeled BPQDs/PLGA NSs determined by measuring the Cy5.5 fluorescence intensity in the blood of the MCF7 tumor bearing Balb/c mice at different time points post-injection of the NSs. The pharmacokinetics obeys a typical two compartment model (as shown by the fitted curve). (b) *In vivo* fluorescence images of the NSs-treated mice at different time points post-injection. (c) *Ex vivo* fluorescence images of the tumor and major organs from the NSs-treated mice at 24 h post-injection. T: tumor, H: heart, Lu: lung, Li: liver, I: intestine, Sp: spleen, K: kidney, and St: stomach. (d) Fluorescence microscopy images of the tumor sections at macro-organizational level (up) and micro-organizational level (down) from the NSs-treated mice. The NSs are shown in red and the nuclei are shown in blue by staining with DAPI. (e) Quantitative biodistribution analysis of the NSs in mice by measuring the Cy5.5 fluorescence intensity in the tumors and major organs at different time points post-injection.

Added in the revised manuscript (Line 18, Page 11 to Line 12, Page 13):

“*In vivo* biodistribution. Since significant amounts of P exist in the animal body, it is very difficult to directly obtain biodistribution information of the BP-based materials (Supplementary Fig. 7). Therefore, in order to study the *in vivo* behavior of the BPQDs/PLGA NSs, Cy5.5, a commonly used NIR fluorescent dye⁶¹ was utilized to label the BPQDs/PLGA NSs by entrapping it into the NSs using the oil-in-water emulsion solvent evaporation method mentioned above. The synthesized Cy5.5-labeled BPQDs/PLGA NSs with similar size of the BPQDs/PLGA NSs exhibit bright and stable fluorescence at about 695 nm (Supplementary Fig. 8) enabling non-invasive monitoring and quantitative examination of the NSs biodistribution in the mice. Hence, the Balb/c nude mice bearing MCF7 breast tumors are intravenously injected with the Cy5.5-labeled BPQDs/PLGA NSs (100 μ L of 1 mg BP/mL for each mouse) for the biodistribution examinations.

The pharmacokinetics profile of the Cy5.5-labeled BPQDs/PLGA NSs was examined by fluorometry to determine the concentrations in blood at different time intervals post-injection (Fig. 6a). Blood circulation of the NSs obeys the typical two compartment model. After the first phase (distribution phase, with a rapid decline) with a half-life of only 1.50 ± 0.21 h, the NSs in circulating blood exhibit a long second phase (elimination phase, the predominant process for NSs clearance) with a half-life of 22.66 ± 3.65 h. The volume of distribution (V) is measured to be 2.31 ± 0.72 mL and the area under curve (AUC) is 0.65 ± 0.11 mg·h/mL. The long blood circulation of the NSs delays the macrophage clearance in reticuloendothelial systems (RES)⁶², favoring enhanced tumor targeting by the EPR effect.

The biodistribution of the Cy5.5-labeled BPQDs/PLGA NSs in the mice is directly observed by fluorescence imaging. As shown in Fig. 6b, considerable fluorescence can be observed from the tumor at 1 h post-injection and the subcutaneous tumor can be definitely delineated from the other tissues. The fluorescence intensity in the tumor gradually increases up to 24 h, indicating that the NSs can continuously accumulate at the tumor site. At 48 h post-injection, the tumor still maintains strong fluorescence, suggesting good retention of the NSs in the tumor. Fig. 6c shows the *ex vivo* fluorescence images obtained at 24 h post-injection, in which bright fluorescence can be observed from the tumor and some organs including the liver, spleen, and kidney. Macro-organizational examination of a tumor (about 50 mm²) in Fig. 6d (up) shows that the NSs are distributed throughout the entire tumor section. Moreover, Fig. 6d highlights the significant colocalization of the nuclei (DAPI staining, shown in blue) and NSs (shown in red) in the tumor section, confirming efficient penetration of the NSs within the

tumor.

A quantitative biodistribution analysis of the Cy5.5-labeled BPQDs/PLGA NSs in mice is conducted (Fig. 6e). The tumor and major organs were collected from the mice, weighed, and solubilized by a lysis buffer at different time intervals post-injection. The homogenized tissue lysates were diluted and measured by fluorometry to quantitatively determine the NSs concentrations. At 24 h post-injection, large NSs concentrations can be found from not only the tumor, but also organs including the liver, spleen, and kidney as consistent with the above *ex vivo* fluorescence examination. Uptake of the NSs by the liver and spleen may be due to RES absorption⁶², while the kidney uptake can be associated with possible renal excretion⁶³. Even so, considerable uptake of the NSs by the tumor can be achieved on account of the EPR effect²¹.”

Added in the “Methods” section (Line 7, Page 18):

“The Cy5.5 NIR fluorescence dye was purchased from Lumiprobe (US).”

Added in the “Methods” section (Line 3, Page 22 to Line 22, Page 23):

“**Pharmacokinetic and biodistribution analysis.** In the biodistribution and pharmacokinetic analysis, fluorescent labeled BPQDs/PLGA NSs were prepared by adding 0.1 mg/mL of Cy5.5 to the BPQDs/PLGA solution in DCM followed by oil-in-water emulsion solvent evaporation described above. The excess dye molecules were removed by centrifugation and washed away with water more than 5 times until no noticeable color change was observed from the supernatant fluids followed by resuspension in PBS.

The female Balb/c nude mice (6 weeks old) were purchased from Slac Laboratory Animal Co.Ltd (Hunan, China). In the pharmacokinetic analysis, blood circulation was assessed by drawing 10 μ L of blood from the tail veins of the Balb/c nude mice at certain time intervals post-injection of the Cy5.5-labeled BPQDs/PLGA NSs. Each blood sample was dissolved in 1 mL of the lysis buffer (the same as the above used) and the concentration of the NSs in the blood was determined from the fluorescence spectrum acquired on a Fluoromax 4 fluorometer (Horiba Jobin Yvon, France). A series of dilutions of the was performed to obtain a standard calibration curve. The blank blood sample without injection was measured to determine the blood auto-fluorescence level, which was subtracted from the fluorescence intensities of injected samples during concentration calculation. The pharmacokinetic

parameters such as half-life ($t_{1/2}$), V , and AUC were determined using a Microsoft add-in tool, PKSolver⁶⁷.

In the *in vivo* fluorescence imaging experiments, the Balb/c nude mice bearing MCF7 breast tumors were intravenously injected with the Cy5.5-labeled BPQDs/PLGA NSs (100 μ L of 1 mg BP/mL for each mouse) and examined by a fluorescence (Xenogen IVIS-Spectrum) imaging system as a function of time for up to 48 h. NIR light with a peak wavelength of 675 nm was used as the excitation source and *in vivo* spectral imaging with the Cy5.5 bandpass emission filter (680 nm to 720 nm) was carried out for an exposure time of 200 ms for each image frame. All the images were captured using identical system settings and auto-fluorescence was removed using the spectral unmixing software.

In the *ex vivo* fluorescence imaging experiments, the NSs-treated mice were sacrificed by cervical dislocation and the corresponding major organs and tissues including the liver, spleen, kidney, heart, stomach, lung, intestine, and tumor were collected and imaged immediately afterwards. The tumors were fixed in 10% neutral buffered formalin and embedded in paraffin. Sections of whole tumor were stained using DAPI (shown in blue) to label all nuclei of tumor cells. The fluorescence images of the tumor sections were acquired on the Leica DM4000B fluorescence microscope (Leica, Nussloch, Germany).

In the quantitative biodistribution analysis, the NSs-treated mice were sacrificed and the organs/tissues were weighed and solubilized by a lysis buffer (1% SDS, 1% Triton X-100, 40 mM Tris Acetate) using a PowerGen homogenizer (Fisher Scientific). The clear homogeneous tissue lysates were diluted 100 times to avoid light scattering and self-quenching during fluorescence measurement. The fluorescence intensities of both the standard samples and real tissues were adjusted to be in the linear range by appropriate dilution and subjected to fluorometry to quantitatively determine the NSs concentrations. The organs and tissues from a control mouse without injection of the NSs were collected and used as controls to subtract the autofluorescence background in various tissues. The samples were measured in triplicate to ensure reproducibility and accuracy. The biodistribution of the NSs in the various organs of the mice was calculated and presented as the percentage of injected dose per gram of tissue (%ID/g).”

Added in the “References” section (References 61-63, 67):

61. Shin, T. H., Choi, Y., Kim, S., & Cheon, J.. Recent advances in magnetic nanoparticle-based multi-modal imaging. *Chem. Soc. Rev.* **44**, 4501-4516 (2015).

62. Huynh, N. T., Roger, E., Lautram, N., Benoît, J. P., & Passirani, C. The rise and rise of stealth nanocarriers for cancer therapy: passive versus active targeting. *Nanomedicine* **5**, 1415-1433 (2010).
63. Yang, K. *et al.* *In vivo* pharmacokinetics, long-term biodistribution, and toxicology of PEGylated graphene in mice. *ACS nano* **5**, 516-522 (2010).
67. Zhang, Y., Huo, M., Zhou, J., & Xie, S. PKSolver: An add-in program for pharmacokinetic and pharmacodynamic data analysis in Microsoft Excel. *Comput. Meth. Prog. Bio.* **99**, 306-314 (2010).

Comment 4: The intratumoral microdistribution of the particles and the intratumoral levels of apoptosis should be addressed, as it could be that the photothermal damage is solely occurring on the blood vessels if the particles do not penetrate within the tumor, resulting in a vascular disruption therapy rather than photothermal ablation.

Reply 3: Per your suggestion, we have performed additional experiments to examine the intratumoral distribution of the BPQDs/PLGA nanospheres to confirm the penetration of the nanospheres within the tumor. In addition, the intratumoral level of apoptosis has been examined. The corresponding results have been added to the revised manuscript.

Added in the revised manuscript (Figure 6d):

Figure 6 | (d) Fluorescence microscopy images of the tumor sections at macro-organizational level (up) and micro-organizational level (down) from the NSs-treated mice. The NSs are shown in red and the nuclei are shown in blue by staining with DAPI.

Added in the revised manuscript (Line 21, Page 12):

“Macro-organizational examination of a tumor (about 50 mm²) in Fig. 6d (up) shows that the NSs are distributed throughout the entire tumor section. Moreover, Fig. 6d highlights the significant colocalization of the nuclei (DAPI staining, shown in blue) and NSs (shown in red) in the tumor section, confirming efficient penetration of the NSs within the tumor.”

Added in the Supplementary Information (Supplementary Figure 12):

Supplementary Figure 12 | Apoptosis detection after *in vivo* photothermal cancer therapy.

Fluorescence microscopy images of the apoptotic cells in the tumor tissues of the Balb/c nude mice irradiated by the 808 nm laser (1 W/cm²) at 24 h after separate intravenous injection with 100 μL of PBS, PLGA NSs, BPQDs (1 mg/mL) and BPQDs/PLGA NSs (1 mg BP/mL). The apoptotic cells are labeled with FITC using the TUNEL assay (shown in green) and the nuclei are stained with DAPI (shown in blue).

Added in the Supplementary Information (Supplementary Figure 13):

Supplementary Figure 13 | Apoptosis examination at the macro-organizational level. Fluorescence microscopy images of the apoptotic cells in the tumor sections at the macro-organizational level (about 10 mm²) acquired from the Balb/c nude mice irradiated by the 808 nm laser (1 W/cm²) at 24 h after intravenous injection with 100 μL of BPQDs/PLGA NSs (1 mg BP/mL). The apoptotic cells are labeled with FITC using the TUNEL assay (shown in green) and the nuclei are stained with DAPI (shown in blue).

Added in the revised manuscript (Line 3, Page 16):

“The anticancer efficiency is further analyzed by a TUNEL (terminal deoxynucleotidyl transferase-mediated deoxyuridine triphosphate nick end-labeling) assay, which is generally utilized to detect the intratumoral levels of apoptosis. As shown in Supplementary Fig. 12, no or few TUNEL-positive cells (shown in green) are observed from the PBS, PLGA, and BPQDs groups, while significant colocalization of nuclei (DAPI staining, shown in blue) and TUNEL-positive apoptotic cells (shown in green) can be observed from the BPQDs/PLGA NSs group. Moreover, the apoptosis examination at macro-organizational level (about 10 mm²) shows that the TUNEL-positive apoptotic cells are distributed throughout the tumor section (Supplementary Fig. 13). These results indicate that the NSs-mediated PTT can induce cancer cell death by activating apoptosis in the tumor.”

Added in the “Methods” section (Line 1, Page 25):

“**Apoptosis detection.** The tumors were collected from the Balb/c nude mice treated with PBS, PLGA NSs, BPQDs and BPQDs/PLGA NSs 24 h after the treatment. The individual tumors were fixed in 10% neutral buffered formalin, embedded in paraffin, sectioned at 5 micrometers, and stained using the TUNEL technique using the *In Situ Cell Death Detection Kit* (Roche Applied Science, Germany). The

experimental procedures were in accordance with the manufacturer's instructions. DAPI was used to stain the sections in the absence of light to label the apoptotic cells and cellular DNA. The fluorescence images were taken on the Leica DM4000B fluorescence microscope (Leica, Nussloch, Germany).”

Comment 5: The *in vivo* study was done against MCF7 tumors for 14 days and the mice showed tumor free survival for 40 days. However, the pictures in figure 5c do not support these observations as the mice treated with the particles still show large tumors. Moreover, MCF7 tumor model may not be appropriate for studying photothermal therapy, as it is prepared in mice with a deficient immune system. Thus, the activity of the particles should be studied in a second model with competent immune system and relevant to photothermal therapy to validate the applicability of the particles.

Reply 5: The original pictures in Figure 5c show the black scar induced by the PTT at the tumor site, which may be mistaken for a “large tumor”. To avoid such misunderstanding, we have prolonged the examination time until the scar at the tumor site is completely cured. Moreover, BALB/c mice with a competent immune system have been added as another animal model (for another major cancer: B16 melanoma tumors) to examine the applicability of the nanospheres-mediated photothermal therapy. Corresponding *in vitro* experiments of B16 melanoma cells are also added.

Changed in the revised manuscript (Figure 7 replacing original Figure 5):

Figure 7 | *In vivo* photothermal cancer therapy. (a) Infrared thermographic maps and (b) Time-dependent temperature increase in the MCF7 breast tumor-bearing nude mice irradiated by the 808 nm laser (1 W/cm^2) at 24 h after separate intravenous injection with $100 \mu\text{L}$ of PBS, PLGA NSs, BPQDs (1 mg/mL) and BPQDs/PLGA NSs (1 mg BP/mL) with the color bar referring to the relative temperature. (c) Typical photographs and (d) Corresponding growth curves of MCF7 breast tumor in different groups of nude mice treated with PBS, PLGA NSs, BPQDs (1 mg/mL) and BPQDs/PLGA NSs (1 mg BP/mL) with the NIR laser irradiation.

Added in the revised manuscript (Line 12, Page 15):

“It is remarkable that for the mice injected with the BPQDs/PLGA NSs and irradiated by the NIR laser, their tumors shrink gradually, are obliterated only leaving black scars, and then are completely cured in about 16 days.”

Changed in the revised manuscript (Figure 4):

Figure 4 | Cell experiments. (a) Relative viability of the HSF normal cells, MCF7 cancer cells, and B16 melanoma cells after incubation with BPQDs/PLGA NSs (internal BPQDs concentrations of 0, 2, 5, 10, 20, 50, and 100 ppm) for 48 h. (b) Relative viability of the MCF7 and B16 cells after incubation with BPQDs/PLGA NSs (internal BPQDs concentrations of 0, 2, 5, 10, and 20 ppm) for 4 h after irradiation with the 808 nm laser (1 W/cm^2) for 10 min. (c) Corresponding fluorescence images of the cells stained with calcein AM (live cells, green fluorescence) and PI (dead cells, red fluorescence).

Added in the Supplementary Information (Supplementary Figure 11):

Supplementary Figure 11 | *In vivo* photothermal cancer therapy in the B16 tumor-bearing mice.

(a) Infrared thermographic maps and (b) Time-dependent temperature increase in the B16 tumor-bearing nude mice irradiated by the 808 nm laser (1 W/cm^2) at 24 h after separate intravenous injection with $100 \mu\text{L}$ of PBS, PLGA NSs, BPQDs (1 mg/mL) and BPQDs/PLGA NSs (1 mg BP/mL) with the color bar referring to the relative temperature. (c) Typical photographs of the B16 tumor-bearing mice and (d) Corresponding growth curves of B16 tumor in different groups of mice treated with $100 \mu\text{L}$ of PBS, PLGA NSs, BPQDs (1 mg/mL), and BPQDs/PLGA NSs (1 mg BP/mL) with the laser irradiation.

Added in the revised manuscript (Line 4, Page 17):

“To further demonstrate the applicability of the NSs-mediated PTT, BALB/c mice, with a competent immune system, were employed as another animal model in the photothermal treatments. As shown in Supplementary Fig. 11, the BPQDs/PLGA NSs also exhibit excellent photothermal efficacy to kill the melanoma tumor in the BALB/c mice without causing obvious toxic side effects. The results demonstrate that the NSs-mediated PTT is suitable for such two kinds of animals with different immune systems.”

Added in the “Methods” section (Line 17, Page 24):

“Another tumor model with competent immune system was established *in situ* in the Balb/c mice by subcutaneously injection of 1×10^7 B16 melanoma cells in PBS to the left rear flank of each mouse. When the tumor volume reached 100 mm^3 , aliquots (100 μL) of PBS, PLGA NSs, BPQDs (1 mg/mL), and BPQDs/PLGA NSs (1 mg BP/mL) were injected separately into the mice *via* the tail vein. At 24 h post-injection, *in vivo* photothermal experiments were conducted as described above.”

Comment 6: Photothermal therapy is limited by the penetration of light in tissues. Therefore, it is mainly applied to superficial tumors. Improving the depth for photothermal damage can result in innovative cancer treatments. Therefore, I recommend the authors to check if these particles can improve the damage in deep tissues.

Reply 6: Per your suggestion, we have conducted additional experiments to examine the depth of photothermal damage induced by the nanospheres. The corresponding results and discussion have been added to the revised manuscript.

Added in the Supplementary Information (Supplementary Figure 14):

Supplementary Figure 14 | Apoptosis detection of tumor sections at different depths. Fluorescence microscopy images of the apoptotic cells in the tumor sections at different depths of the Balb/c nude mice irradiated by the 808 nm laser (1 W/cm^2) at 24 h after intravenous injection with 100 μL BPQDs/PLGA NSs (1 mg BP/mL). The apoptotic cells are labeled with FITC using the TUNEL assays (shown in green) and the nuclei are stained with DAPI (shown in blue).

Added in the revised manuscript (Line 12, Page 16):

“The depth of photothermal damage is investigated in the tumor bearing nude mice with the tumor volume as large as 1000 mm^3 . After the photothermal treatment, intratumoral apoptosis of the tumor sections at different depths was detected by a TUNEL assay (Supplementary Fig. 14). Most cancer cells undergo apoptosis in the tumor sections at depths of no more than 6 mm. Although evident depth-dependent decay of the PTT efficiency is observed when the depth is over 6 mm, significant apoptosis of cells can still be found from the section at the depth of 10 mm. The considerable photothermal damage to deep tissues stems from the excellent PTT efficiency of the NSs and high tissue penetration ability of NIR light. It should be noted that although the penetration depth of NIR light is limited to be no more than 10 mm^2 , clinical photothermal treatment of deep tumors is still achievable

with the aid of specialized medical devices such as endoscopic ones in combination with optical fibers as well as implanted NIR devices^{65,66}.”

Added in the “Methods” section (Line 7, Page 25):

“To further investigate the depth of the photothermal damage, the tumor bearing nude mice with a tumor volume as large as 1000 mm³ was illuminated with the 808 nm NIR laser (1 W/cm²) for 10 min. Intratumoral apoptosis of the tumor sections was monitored at different depths (2, 4, 6, 8 and 10 mm) by the TUNEL assay.”

Added in the “References” section (References 65-66):

65. Jaque, D. *et al.* Nanoparticles for photothermal therapies. *Nanoscale* **6**, 9494-9530 (2014).
66. Bagley, A. F., Hill, S., Rogers, G. S., & Bhatia, S. N. Plasmonic photothermal heating of intraperitoneal tumors through the use of an implanted near-infrared source. *ACS Nano* **7**, 8089-8097 (2013).

Comment 7: The toxicity from the particles was completely ignored. The authors should demonstrate the safety of these particles without irradiation, with irradiation, and by exposing the mice to daylight (as a main issue for photosensitizer molecules is the side effects arising after sun exposure).

Reply 7: We have examined the *in vivo* toxicity of the nanospheres by performing the hematological, blood biochemical, and histological analyses. According to your suggestion, the mice in these examinations have been randomly divided into 4 groups and subjected to variable conditions, including: (1) Control group without any treatment, (2) NSs directly intravenously injected into the mice, (3) NSs intravenously injected into the mice after 808 nm laser irradiation for 10 min, and (4) NSs intravenously injected into the mice which are then exposed to artificial daylight for 24 h. The corresponding results and discussion have been added to the revised manuscript.

Added in the revised manuscript (Figure 5):

Figure 5 | *In vivo* toxicity. (a) Hematological data of the mice intravenously injected with the BPQDs/PLGA NSs at 1, 7, and 28 days post-injection. The terms are following: white blood cells (WBC), red blood cells (RBC), hemoglobin (HGB), mean corpuscular volume (MCV), mean corpuscular hemoglobin (MCH), mean corpuscular hemoglobin concentration (MCHC), platelets (PLT), and hematocrit (HCT). (b) Blood biochemical analysis of the NSs-treated mice at 1, 7, and 28 days post-injection. The results show the mean and standard deviation of aminotransferase (ALT), aminotransferase (AST), total protein (TP), globulin (GLB), total bilirubin (TBIL), blood urea nitrogen (BUN), creatinine (CREA), and albumin (AL B). (c) Histological data (H&E stained images) obtained from the liver, spleen, kidney, heart, and lung of the NSs-treated mice at 1, 7, and 28 days post-injection.

Added in the revised manuscript (Line 9, Page 10 to Line 17, Page 11):

“*In vivo* toxicity. The *in vivo* toxicology of the BPQDs/PLGA NSs is investigated systematically. Sixty healthy female Balb/c mice (6 weeks old) were randomly divided into 4 groups and subjected to variable conditions, including: (1) Control group without any treatment, (2) NSs directly intravenously injected into the mice, (3) NSs intravenously injected into the mice after 808 nm laser irradiation for 10 min, and (4) NSs intravenously injected into the mice which are then exposed to artificial daylight for 24 h. The injection dose of the NSs is about 10 mg BP/kg and hematological, blood biochemical, and histological analyses were performed at time points of 1, 7, and 28 days post-injection.

The standard hematology markers including the white blood cells (WBC), red blood cells (RBC), hemoglobin (HGB), mean corpuscular volume (MCV), mean corpuscular hemoglobin (MCH), mean corpuscular hemoglobin concentration (MCHC), platelets (PLT), and hematocrit (HCT) were measured (Fig. 5a). Compared to the control group, all the parameters in the three NSs-treated groups at all time points appear to be normal and the differences between are not statistically significant (P value > 0.05). These results indicate that the BPQDs/PLGA NSs do not cause obvious infection and inflammation in the treated mice⁵⁹.

Blood biochemical analyses were carried out and various parameters including alanine transaminase (ALT), aspartate transaminase (AST), total protein (TP), globulin (GLB), total bilirubin (TBIL), blood urea nitrogen (BUN), creatinine (CREA), and albumin (ALB) were examined (Fig. 5b). Compared to the control group, no meaningful difference can be observed from the three NSs-treated groups at all time points. Hence, the NSs treatment does not affect the blood chemistry of mice. Furthermore, since ALT, AST, and CREA are closely related to the functions of the liver and kidney of mice⁵⁹, the results demonstrate that the NSs induce no obvious hepatic and kidney toxicity in mice.

Finally, the corresponding histological changes of organs were checked by immunohistochemistry using major organs including the liver, spleen, kidney, heart, and lung collected and sliced for hematoxylin and eosin (H&E) staining (Fig. 5c). No noticeable signal of organ damage can be observed during the whole treatment period from all the groups suggesting no apparent histological abnormalities or lesions in the NSs-treated groups for the test dose.

According to above analyses, inappreciable toxicity is observed from the BPQDs/PLGA NSs regardless of NIR laser irradiation. Even if the NSs-treated mice are under artificial daylight illumination for 24 h, no significant toxic side effects can be found, indicating that the NSs induce no

evident phototoxicity which has generally been observed from many photosensitizer molecules⁶⁰. These results demonstrate the good biocompatibility of the BPQDs/PLGA NSs.”

Added in the “Methods” section (Line 11, Page 21 to Line 2, Page 22):

“***In vivo* toxicity.** Sixty healthy female Balb/c mice (6 weeks old) were obtained from Slac Laboratory Animal Co.Ltd (Hunan, China) and all the *in vivo* experiments followed the protocols approved by the Animal Care and Use Committee of the Shenzhen Institutes of Advanced Technology, Chinese Academy of Sciences. Aliquots (200 μ L) of the BPQDs/PLGA NSs (1 mg BP/mL) were injected separately into the mice *via* the tail vein. The mice were randomly divided into 4 groups and subjected to variable conditions. This include: (1) Control group without any treatment, (2) NSs directly intravenously injected into the mice, (3) NSs intravenously injected into the mice after 808 nm laser irradiation for 10 min, and (4) NSs intravenously injected into the mice which are then exposed to artificial daylight for 24 h. The mice were sacrificed at various time points after injection (1, 7, and 28 days, five mice per group at each time point). About 0.8 mL of blood were collected from each mouse to conduct complete blood panel analysis and serum biochemistry assay at the Shanghai Research Center for Biomodel Organism. The major organs (liver, spleen, kidney, heart, and lung) were harvested, fixed in 10% neutral buffered formalin, processed routinely into paraffin, sectioned at 8 μ m, stained with H&E, and examined by digital microscopy.”

Added in the “References” section (References 59,60):

59. Zhang, X. D. *et al.* Metabolizable Bi₂Se₃ nanoplates: biodistribution, toxicity, and uses for cancer radiation therapy and imaging. *Adv. Funct. Mater.* **24**, 1718-1729 (2014).
60. Allison, R. R. *et al.* Photosensitizers in clinical PDT. *Photodiagn. Photodyn. Ther.* **1**, 27-42 (2004).

Comment 8: The authors mentioned that the higher efficacy of their particles over the BPQDs is because the 100 nm size of the nanoparticles is more appropriate for efficient tumor targeting compared to the ultrasmall size of BPQDs (3 nm). However, this observation is biased from the experimental setting. Accordingly, the dose and time for irradiation (24 h after injection) of the antitumor experiment were more suitable for the nanoparticles than for the BPQDs. If the experiment is performed with higher BPQDs doses and earlier irradiations, the results could provide better antitumor effect for BPQDs than

for PLGA/BPQDs, making the development of PLGA/BPQDs unnecessary. The authors should demonstrate that at the optimal conditions for each formulation, i.e. dose and irradiation timing for BPQDs and PLGA/BPQDs, the PLGA/BPQDs still perform better than BPQDs.

Reply 8: Per your suggestion, we have examined the time-dependent temperature increase in the tumor-bearing nude mice after separate injection of the BPQDs and BPQDs/PLGA NSs with different concentrations (0.5, 1.0, 2.0 and 3.0 mg BP/mL) and irradiated with the NIR laser at different times (1, 4, 8, 12, 24 and 48 h) post-injection. The corresponding results have been added to the revised manuscript.

Added in the Supplementary Information (Supplementary Figure 10):

Supplementary Figure 10 | Photothermal performance comparison between the BPQDs/PLGA NSs and bare BPQDs for *in vivo* tumor ablation. (a, b) Infrared thermographic maps and (c, d) Time-dependent temperature increase in the tumor-bearing nude mice after separate intravenous injection of the BPQDs and BPQDs/PLGA NSs with different concentrations (0.5, 1, 2 and 3 mg

BP/mL) and irradiation with the 808 nm laser (1 W/cm^2) at different time points (1, 4, 8, 12, 24 and 48 h) after injection. The color bars refer to the relative temperature.

Added in the revised manuscript (Line 17, Page 14 to Line 6, Page 15):

“To further evaluate the influence of PLGA encapsulation on the *in vivo* PTT efficiency of the BPQDs, the BPQDs and BPQDs/PLGA NSs with different concentrations (0.5, 1.0, 2.0 and 3.0 mg BP/mL) were injected intravenously into the tumor-bearing nude mice, which were irradiated with the NIR laser at different times (1, 4, 8, 12, 24 and 48 h) post-injection. As shown in Supplementary Fig. 10, the BPQDs/PLGA NSs produce larger tumor temperature increase than the bare BPQDs under all conditions. The better *in vivo* PTT efficiency of the BPQDs/PLGA NSs than bare BPQDs pertaining to tumor ablation can be attributed to two factors. Firstly, the BPQDs/PLGA NSs have better stability than the bare BPQDs and so can maintain the photothermal performance during circulation in the body. Secondly, compared to the ultrasmall BPQDs with a size of about 3 nm^{16} , the size of the BPQDs/PLGA NSs of about 100 nm is more appropriate for efficient tumor targeting and retention during the long blood circulation in the body.”

Added in the “Methods” section (Line 12, Page 24):

“To further compare the photothermal effects between the BPQDs and BPQDs/PLGA NSs in details, photothermal experiments with different injection concentrations (0.5, 1, 2 and 3 mg BP/mL) and irradiation time (1, 4, 8, 12, 24 and 48 h post-injection) were performed. The temperature of the tumors and infrared thermographic maps were obtained by the infrared thermal imaging camera.”

Comment 9: English should be carefully revised. There are too many misspellings throughout the manuscript.

Reply 9: We have revised the manuscript carefully to correct misspelling.

Replies to the 3rd reviewer's comments (NCOMMS-16-03858)

Comment 1: The data cannot support the conclusions very well, especially the excellent tumor targeting ability and biocompatibility of BPQDs/PLGA nanosphere in vivo. In addition, the in vivo studies on the unique biodegradability and excellent biocompatibility of BPQDs/PLGA nanosphere are important for its further application. Therefore, the distribution and metabolism of BPQDs/PLGA nanosphere in mice should be investigated.

Reply 1: Thank you for your suggestion. We have performed additional experiments to examine the distribution and metabolism of the BPQDs/PLGA nanospheres. The corresponding results have been added to the revised manuscript.

Added in the Supplementary Information (Supplementary Figure 8):

Supplementary Figure 8 | Characterization of the Cy5.5-labeled BPQDs/PLGA NSs. (a) SEM image of the Cy5.5-labeled BPQDs/PLGA NSs. (b) Photographs of the BPQDs/PLGA NSs and Cy5.5-labeled BPQDs/PLGA NSs. (c) Emission spectra ($\lambda_{\text{ex}} = 675 \text{ nm}$) of the Cy5.5-labeled BPQDs/PLGA NSs before and after storage in PBS for 7 days. No fluorescence can be observed from the BPQDs/PLGA NSs without Cy5.5 labeling.

Added in the revised manuscript (Figure 6):

Figure 6 | Pharmacokinetic and biodistribution analysis. (a) Blood circulation curve of the Cy5.5-labeled BPQDs/PLGA NSs determined by measuring the Cy5.5 fluorescence intensity in the blood of the MCF7 tumor bearing Balb/c mice at different time points post-injection of the NSs. The pharmacokinetics obeys a typical two compartment model (as shown by the fitted curve). (b) *In vivo* fluorescence images of the NSs-treated mice at different time points post-injection. (c) *Ex vivo* fluorescence images of the tumor and major organs from the NSs-treated mice at 24 h post-injection. T: tumor, H: heart, Lu: lung, Li: liver, I: intestine, Sp: spleen, K: kidney, and St: stomach. (d) Fluorescence microscopy images of the tumor sections at macro-organizational level (up) and micro-organizational level (down) from the NSs-treated mice. The NSs are shown in red and the nuclei are shown in blue by staining with DAPI. (e) Quantitative biodistribution analysis of the NSs in mice by measuring the Cy5.5 fluorescence intensity in the tumors and major organs at different time points post-injection.

Added in the revised manuscript (Line 18, Page 11 to Line 22, Page 13):

“***In vivo* biodistribution.** Since significant amounts of P exist in the animal body, it is very difficult to directly obtain biodistribution information of the BP-based materials (Supplementary Fig. 7). Therefore, in order to study the *in vivo* behavior of the BPQDs/PLGA NSs, Cy5.5, a commonly used NIR fluorescent dye⁶¹ was utilized to label the BPQDs/PLGA NSs by entrapping it into the NSs using the oil-in-water emulsion solvent evaporation method mentioned above. The synthesized Cy5.5-labeled

BPQDs/PLGA NSs with similar size of the BPQDs/PLGA NSs exhibit bright and stable fluorescence at about 695 nm (Supplementary Fig. 8) enabling non-invasive monitoring and quantitative examination of the NSs biodistribution in the mice. Hence, the Balb/c nude mice bearing MCF7 breast tumors are intravenously injected with the Cy5.5-labeled BPQDs/PLGA NSs (100 μ L of 1 mg BP/mL for each mouse) for the biodistribution examinations.

The pharmacokinetics profile of the Cy5.5-labeled BPQDs/PLGA NSs was examined by fluorometry to determine the concentrations in blood at different time intervals post-injection (Fig. 6a). Blood circulation of the NSs obeys the typical two compartment model. After the first phase (distribution phase, with a rapid decline) with a half-life of only 1.50 ± 0.21 h, the NSs in circulating blood exhibit a long second phase (elimination phase, the predominant process for NSs clearance) with a half-life of 22.66 ± 3.65 h. The volume of distribution (V) is measured to be 2.31 ± 0.72 mL and the area under curve (AUC) is 0.65 ± 0.11 mg·h/mL. The long blood circulation of the NSs delays the macrophage clearance in reticuloendothelial systems (RES)⁶², favoring enhanced tumor targeting by the EPR effect.

The biodistribution of the Cy5.5-labeled BPQDs/PLGA NSs in the mice is directly observed by fluorescence imaging. As shown in Fig. 6b, considerable fluorescence can be observed from the tumor at 1 h post-injection and the subcutaneous tumor can be definitely delineated from the other tissues. The fluorescence intensity in the tumor gradually increases up to 24 h, indicating that the NSs can continuously accumulate at the tumor site. At 48 h post injection, the tumor still maintains strong fluorescence, suggesting good retention of the NSs in the tumor. Fig. 6c shows the *ex vivo* fluorescence images obtained at 24 h post-injection, in which bright fluorescence can be observed from the tumor and some organs including the liver, spleen, and kidney. Macro-organizational examination of a tumor (about 50 mm²) in Fig. 6d (up) shows that the NSs are distributed throughout the entire tumor section. Moreover, Fig. 6d highlights the significant colocalization of the nuclei (DAPI staining, shown in blue) and NSs (shown in red) in the tumor section, confirming efficient penetration of the NSs within the tumor.

A quantitative biodistribution analysis of the Cy5.5-labeled BPQDs/PLGA NSs in mice is conducted (Fig. 6e). The tumor and major organs were collected from the mice, weighed, and solubilized by a lysis buffer at different time intervals post-injection. The homogenized tissue lysates were diluted and measured by fluorometry to quantitatively determine the NSs concentrations. At 24 h post-injection, large NSs concentrations can be found from not only the tumor, but also organs including the liver,

spleen, and kidney as consistent with the above *ex vivo* fluorescence examination. Uptake of the NSs by the liver and spleen may be due to RES absorption⁶², while the kidney uptake can be associated with possible renal excretion⁶³. Even so, considerable uptake of the NSs by the tumor can be achieved on account of the EPR effect²¹.

Since the NSs in the physiological medium can maintain their integrity (Fig. 3c,d) without causing evident fluorescence decrease of the entrapped Cy5.5 (Supplementary Fig. 8) for 7 days, the fluorescence examinations were further used to estimate the time-dependent residual amounts of the NSs in mice during the 7 days post-injection. The residual ratios were calculated by normalizing the total residual amounts in these organs and tissues to initial total amounts. It can be calculated that the residual ratio of the NSs decreases from 90.1%ID/g at day 1 (24 h) to only 29.9%ID/g at day 7, suggesting the possibility of the NSs to be partially metabolized. It is known that such nanoparticles is generally difficult to be completely metabolized and excreted from the body directly. However, the aforementioned biodegradability of the NSs enables harmless clearance from the body in a reasonable period of time (for example, several months).”

Added in the “Methods” section (Line 7, Page 18):

“The Cy5.5 NIR fluorescence dye was purchased from Lumiprobe (US).”

Added in the “Methods” section (Line 3, Page 22 to Line 22, Page 23):

“Pharmacokinetic and biodistribution analysis. In the biodistribution and pharmacokinetic analysis, fluorescent labeled BPQDs/PLGA NSs were prepared by adding 0.1 mg/mL of Cy5.5 to the BPQDs/PLGA solution in DCM followed by oil-in-water emulsion solvent evaporation described above. The excess dye molecules were removed by centrifugation and washed away with water more than 5 times until no noticeable color change was observed from the supernatant fluids followed by resuspension in PBS.

The female Balb/c nude mice (6 weeks old) were purchased from Slac Laboratory Animal Co.Ltd (Hunan, China). In the pharmacokinetic analysis, blood circulation was assessed by drawing 10 μ L of blood from the tail veins of the Balb/c nude mice at certain time intervals post-injection of the Cy5.5-labeled BPQDs/PLGA NSs. Each blood sample was dissolved in 1 mL of the lysis buffer (the same as the above used) and the concentration of the NSs in the blood was determined from the fluorescence spectrum acquired on a Fluoromax 4 fluorometer (Horiba Jobin Yvon, France). A series of

dilutions of the was performed to obtain a standard calibration curve. The blank blood sample without injection was measured to determine the blood auto-fluorescence level, which was subtracted from the fluorescence intensities of injected samples during concentration calculation. The pharmacokinetic parameters such as half-life ($t_{1/2}$), V , and AUC were determined using a Microsoft add-in tool, PKSolver⁶⁷.

In the *in vivo* fluorescence imaging experiments, the Balb/c nude mice bearing MCF7 breast tumors were intravenously injected with the Cy5.5-labeled BPQDs/PLGA NSs (100 μ L of 1 mg BP/mL for each mouse) and examined by a fluorescence (Xenogen IVIS-Spectrum) imaging system as a function of time for up to 48 h. NIR light with a peak wavelength of 675 nm was used as the excitation source and *in vivo* spectral imaging with the Cy5.5 bandpass emission filter (680 nm to 720 nm) was carried out for an exposure time of 200 ms for each image frame. All the images were captured using identical system settings and auto-fluorescence was removed using the spectral unmixing software.

In the *ex vivo* fluorescence imaging experiments, the NSs-treated mice were sacrificed by cervical dislocation and the corresponding major organs and tissues including the liver, spleen, kidney, heart, stomach, lung, intestine, and tumor were collected and imaged immediately afterwards. The tumors were fixed in 10% neutral buffered formalin and embedded in paraffin. Sections of whole tumor were stained using DAPI (shown in blue) to label all nuclei of tumor cells. The fluorescence images of the tumor sections were acquired on the Leica DM4000B fluorescence microscope (Leica, Nussloch, Germany).

In the quantitative biodistribution analysis, the NSs-treated mice were sacrificed and the organs/tissues were weighed and solubilized by a lysis buffer (1% SDS, 1% Triton X-100, 40 mM Tris Acetate) using a PowerGen homogenizer (Fisher Scientific). The clear homogeneous tissue lysates were diluted 100 times to avoid light scattering and self-quenching during fluorescence measurement. The fluorescence intensities of both the standard samples and real tissues were adjusted to be in the linear range by appropriate dilution and subjected to fluorometry to quantitatively determine the NSs concentrations. The organs and tissues from a control mouse without injection of the NSs were collected and used as controls to subtract the autofluorescence background in various tissues. The samples were measured in triplicate to ensure reproducibility and accuracy. The biodistribution of the NSs in the various organs of the mice was calculated and presented as the percentage of injected dose per gram of tissue (%ID/g).”

Added in the “References” section (References 61-63, 67):

61. Shin, T. H., Choi, Y., Kim, S., & Cheon, J.. Recent advances in magnetic nanoparticle-based multi-modal imaging. *Chem. Soc. Rev.* **44**, 4501-4516 (2015).
62. Huynh, N. T., Roger, E., Lautram, N., Benoît, J. P., & Passirani, C. The rise and rise of stealth nanocarriers for cancer therapy: passive versus active targeting. *Nanomedicine* **5**, 1415-1433 (2010).
63. Yang, K. *et al.* *In vivo* pharmacokinetics, long-term biodistribution, and toxicology of PEGylated graphene in mice. *ACS nano* **5**, 516-522 (2010).
67. Zhang, Y., Huo, M., Zhou, J., & Xie, S. PKSolver: An add-in program for pharmacokinetic and pharmacodynamic data analysis in Microsoft Excel. *Comput. Meth. Prog. Bio.* **99**, 306-314 (2010).

Comment 2: Although BPQDs have a broad absorption across the ultraviolet (UV) and infrared regions, the common PPT agents, such as gold nanorods and carbon nanotube, may have higher photothermal conversion efficiency in NIR, because of their specific absorption peak.

Reply 2: In previous literature (*Angew. Chem.* **127**, 6279-6283 (2015)), it has been demonstrated that the photothermal conversion efficiency of the BPQDs is 28.4 %, which is higher than that of gold nanorods (AuNRs, 21 %). In the revised manuscript, we have provided additional experimental results to compare the *in vitro* and *in vivo* photothermal performance of the BPQDs/PLGA NSs and AuNRs.

Added in the Supplementary Information (Supplementary Figure 6):

Supplementary Figure 6 | Comparison of photothermal performance between the BPQDs/PLGA NSs and AuNRs. (a) Absorption spectra and (b) Photothermal heating curves of the BPQDs/PLGA NSs (20.0 ppm BPQDs) and AuNRs (72.4 ppm) with the same absorption at 808 nm. The AuNRs as the positive control are synthesized in an aqueous solution by a common seed-mediated growth method reported previously^{1,2} and BSA is used to conjugate with the AuNRs using a method described previously^{2,3}. (c) Relative viability of the MCF7 cells after incubation with BPQDs/PLGA NSs and AuNRs with different concentrations (same absorption at 808 nm) for 4 h and irradiated with the 808 nm laser (1 W/cm^2) for 10 min. (d) Corresponding fluorescence images of the cells stained with calcein AM (live cells, green fluorescence) and PI (dead cells, red fluorescence).

Added in the revised manuscript (Line 2, Page 10):

“In the next step, the PTT efficiency of the BPQDs/PLGA NSs was compared with that of gold nanorods (AuNRs), one of the common photothermal agents. On account of the large NIR extinction coefficient and high photothermal conversion efficiency of the BPQDs³⁹, the BPQDs/PLGA NSs are more efficient in increasing the solution temperature than the AuNRs (Supplementary Fig. 6). In the cell photothermal experiments, both of the NSs (containing only 10 ppm of BPQDs) and AuNRs (72.4 ppm)

can kill the cancer cells almost completely, but it is clear that less BPQDs are needed. These results confirm the suitability of BPQDs/PLGA NSs as an efficient PTT agent.”

Added in the Supplementary Information (Supplementary Figure 9):

Supplementary Figure 9 | Photothermal performance comparison of the BPQDs/PLGA NSs and AuNRs for *in vivo* tumor ablation. Infrared thermographic maps of the tumor-bearing nude mice irradiated by the 808 nm laser (1 W/cm^2) at 24 h after separate intravenous injection with 100 μL of BPQDs/PLGA NSs (1 mg BP/mL) and AuNRs (3.62 mg/mL) with the color bar referring to the relative temperature.

Added in the revised manuscript (Line 13, Page 14):

“The AuNRs were employed as a positive control in the photothermal experiments (Supplementary Fig. 9). Under the same irradiation condition, the tumor temperature of the mice injected with 100 μL of the AuNRs (3.62 mg/mL) increases to 54.4 $^{\circ}\text{C}$, which is lower than that induced by the NSs. These results indicate the high efficiency of the BPQDs/PLGA NSs as a PTT agent for *in vivo* tumor ablation.”

Added in the “Methods” section (Line 7, Page 21):

“The AuNRs were employed as a positive control in the *in vitro* photothermal experiments. The MCF7 cells was incubated with AuNRs (concentrations of 0, 7.2, 18.1, 36.2, and 72.4 ppm) for 4 h at 37 $^{\circ}\text{C}$ and then irradiated with the 808 nm laser (1 W/cm^2) for 10 min. The corresponding fluorescence images of the cells and cell viability were assessed by the above method.”

Added in the “Methods” section (Line 8, Page 24):

“The AuNRs (100 μL , 3.62 mg/mL) were employed as a positive control in the photothermal

experiments.”

Added in the Supplementary Information (Supplementary References 1-3, Page 16):

1. Shao, J. *et al.* PLLA nanofibrous paper-based plasmonic substrate with tailored hydrophilicity for focusing SERS detection. *ACS Appl. Mater. Interfaces* **7**, 5391-5399 (2015).
2. Li, Z. *et al.* Small gold nanorods laden macrophages for enhanced tumor coverage in photothermal therapy. *Biomaterials* **74**, 144-154 (2016).
3. Ali, M. R. K., Panikkanvalappil, S. R., & El-Sayed, M. A. Enhancing the efficiency of gold nanoparticles treatment of cancer by increasing their rate of endocytosis and cell accumulation using rifampicin. *J. Am. Chem. Soc.* **136**, 4464-4467 (2014).

Comment 3: The concentrations of BPQDs/PLGA nanosphere and the laser doses in Fig. 2-3 are not clear.

Reply 3: Per your suggestion, we have added the concentrations of the BPQDs/PLGA nanospheres and the laser doses in the “Figure legends” of Fig. 2-3.

Changed in the “Figure legends” (Figure 2):

“**Figure 2 | Stability evaluation under ambient conditions.** (a) Photographs and (b, c) Absorption spectra of the BPQDs and BPQDs/PLGA NSs with the same amount of BPQDs (20 ppm) after storing in water for different periods of time. Insets in (b, c): Tyndall effect (lower left corner) and variation of the absorption ratios (A/A_0) at 808 nm (middle top). (d, e) Raman scattering spectra acquired from the BPQDs and BPQDs/PLGA NSs, respectively, after storing in water for 0 and 8 days. (f, g) Photothermal heating curves of the BPQDs and BPQDs/PLGA NSs, respectively, after storing in water for different periods of time and being irradiated with the 808 nm laser (1 W/cm^2) for 10 min.”

Changed in the “Figure legends” (Figure 3):

“**Figure 3 | Biodegradation performance.** (a) Absorbance spectra of the BPQDs/PLGA NSs (internal BPQDs concentration is 10 ppm) dispersed in PBS for 0 h, 24 h, and 8 weeks with the inset showing the corresponding photographs. (b) Photothermal heating curves of the BPQDs/PLGA NSs dispersed in PBS for 0 h, 24 h, and 8 weeks and irradiated with the 808 nm laser (1 W/cm^2) for 10 min. (c) Residual weight of the BPQDs/PLGA NSs after degradation in PBS as a function of time ($n=5$; $*P < 0.05$, $**P <$

0.01, *** $P < 0.001$; ANOVA). (d) SEM images of the BPQDs/PLGA NSs after degradation in PBS for 1 week, 4 weeks, and 8 weeks together with the corresponding TEM image of the NSs after degradation for 8 weeks. (e) Schematic representation of the degradation process of the BPQDs/PLGA NSs in the physiological environment.”

Comment 4: The BPQDs/PLGA nanosphere may mainly accumulate in liver after i.v. injection into mice, therefore, the toxic analysis in liver and other organs should be determined.

Reply 4: Per your suggestion, we have examined the *in vivo* toxicity of the BPQDs/PLGA nanospheres by performing the hematological, blood biochemical, and histological analyses. The results have been added to the revised manuscript.

Added in the revised manuscript (Figure 5):

Figure 5 | *In vivo* toxicity. (a) Hematological data of the mice intravenously injected with the BPQDs/PLGA NSs at 1, 7, and 28 days post-injection. The terms are following: white blood cells (WBC), red blood cells (RBC), hemoglobin (HGB), mean corpuscular volume (MCV), mean corpuscular hemoglobin (MCH), mean corpuscular hemoglobin concentration (MCHC), platelets (PLT), and hematocrit (HCT). (b) Blood biochemical analysis of the NSs-treated mice at 1, 7, and 28 days post-injection. The results show the mean and standard deviation of aminotransferase (ALT), aminotransferase (AST), total protein (TP), globulin (GLB), total bilirubin (TBIL), blood urea nitrogen (BUN), creatinine (CREA), and albumin (AL B). (c) Histological data (H&E stained images) obtained from the liver, spleen, kidney, heart, and lung of the NSs-treated mice at 1, 7, and 28 days post-injection.

Added in the revised manuscript (Line 9, Page 10 to Line 17, Page 11):

“*In vivo* toxicity. The *in vivo* toxicology of the BPQDs/PLGA NSs is investigated systematically. Sixty healthy female Balb/c mice (6 weeks old) were randomly divided into 4 groups and subjected to variable conditions, including: (1) Control group without any treatment, (2) NSs directly intravenously injected into the mice, (3) NSs intravenously injected into the mice after 808 nm laser irradiation for 10 min, and (4) NSs intravenously injected into the mice which are then exposed to artificial daylight for 24 h. The injection dose of the NSs is about 10 mg BP/kg and hematological, blood biochemical, and histological analyses were performed at time points of 1, 7, and 28 days post-injection.

The standard hematology markers including the white blood cells (WBC), red blood cells (RBC), hemoglobin (HGB), mean corpuscular volume (MCV), mean corpuscular hemoglobin (MCH), mean corpuscular hemoglobin concentration (MCHC), platelets (PLT), and hematocrit (HCT) were measured (Fig. 5a). Compared to the control group, all the parameters in the three NSs-treated groups at all time points appear to be normal and the differences between are not statistically significant (P value > 0.05). These results indicate that the BPQDs/PLGA NSs do not cause obvious infection and inflammation in the treated mice⁵⁹.

Blood biochemical analyses were carried out and various parameters including alanine transaminase (ALT), aspartate transaminase (AST), total protein (TP), globulin (GLB), total bilirubin (TBIL), blood urea nitrogen (BUN), creatinine (CREA), and albumin (ALB) were examined (Fig. 5b). Compared to the control group, no meaningful difference can be observed from the three NSs-treated groups at all time points. Hence, the NSs treatment does not affect the blood chemistry of mice. Furthermore, since

ALT, AST, and CREA are closely related to the functions of the liver and kidney of mice⁵⁹, the results demonstrate that the NSs induce no obvious hepatic and kidney toxicity in mice.

Finally, the corresponding histological changes of organs were checked by immunohistochemistry using major organs including the liver, spleen, kidney, heart, and lung collected and sliced for hematoxylin and eosin (H&E) staining (Fig. 5c). No noticeable signal of organ damage can be observed during the whole treatment period from all the groups suggesting no apparent histological abnormalities or lesions in the NSs-treated groups for the test dose.

According to above analyses, inappreciable toxicity is observed from the BPQDs/PLGA NSs regardless of NIR laser irradiation. Even if the NSs-treated mice are under artificial daylight illumination for 24 h, no significant toxic side effects can be found, indicating that the NSs induce no evident phototoxicity which has generally been observed from many photosensitizer molecules⁶⁰. These results demonstrate the good biocompatibility of the BPQDs/PLGA NSs.”

Added in the “Methods” section (Line 11, Page 21 to Line 2, Page 22):

“*In vivo* toxicity. Sixty healthy female Balb/c mice (6 weeks old) were obtained from Slac Laboratory Animal Co.Ltd (Hunan, China) and all the *in vivo* experiments followed the protocols approved by the Animal Care and Use Committee of the Shenzhen Institutes of Advanced Technology, Chinese Academy of Sciences. Aliquots (200 μ L) of the BPQDs/PLGA NSs (1 mg BP/mL) were injected separately into the mice *via* the tail vein. The mice were randomly divided into 4 groups and subjected to variable conditions. This include: (1) Control group without any treatment, (2) NSs directly intravenously injected into the mice, (3) NSs intravenously injected into the mice after 808 nm laser irradiation for 10 min, and (4) NSs intravenously injected into the mice which are then exposed to artificial daylight for 24 h. The mice were sacrificed at various time points after injection (1, 7, and 28 days, five mice per group at each time point). About 0.8 mL of blood were collected from each mouse to conduct complete blood panel analysis and serum biochemistry assay at the Shanghai Research Center for Biomodel Organism. The major organs (liver, spleen, kidney, heart, and lung) were harvested, fixed in 10% neutral buffered formalin, processed routinely into paraffin, sectioned at 8 μ m, stained with H&E, and examined by digital microscopy.”

Added in the “References” section (References 59,60):

59. Zhang, X. D. *et al.* Metabolizable Bi₂Se₃ nanoplates: biodistribution, toxicity, and uses for cancer radiation therapy and imaging. *Adv. Funct. Mater.* **24**, 1718-1729 (2014).
60. Allison, R. R. *et al.* Photosensitizers in clinical PDT. *Photodiagn. Photodyn. Ther.* **1**, 27-42 (2004).

Comment 5: BPQDs/PLGA nanosphere were used as photothermal agent, therefore, the stability or biodegradation after the laser treatment should be shown to further understand the metabolism of BPQDs/PLGA nanosphere in vivo.

Reply 5: Per your suggestion, we have examined the biodegradation properties of the BPQDs/PLGA nanospheres after the laser treatment. The corresponding results have been added to the revised manuscript.

Added in the Supplementary Information (Supplementary Figure 5):

Supplementary Figure 5 | Biodegradation performance in PBS of BPQDs/PLGA NSs after irradiation by the 808 nm laser (1 W/cm²) for 10 min. (a) Residual weight of the NSs irradiated and degraded in PBS as a function of time (n=5; **P* < 0.05, ***P* < 0.01, ****P* < 0.001; ANOVA). (b) SEM images of the NSs irradiated and then degraded in PBS for 0, 1, 4, and 8 weeks.

Added in the revised manuscript (Line 16, Page 8):

“The influence of laser irradiation on the degradation of the BPQDs/PLGA NSs is further assessed (Supplementary Fig. 5). After laser illumination (808 nm, 1 W/cm²) for 10 min, no evident morphological change and influence on the degradation rate can be found from the NSs suggesting that the photothermal treatment does not affect the biodegradability of the NSs significantly.”

Added in the “Methods” section (Line 4, Page 20):

“Degradation of the BPQDs/PLGA NSs after 808 nm laser (1 W/cm²) illumination for 10 min was also evaluated by the above method.”

Comment 6: The laser treatment should be labeled in the figures 4-5.

Reply 6: We have labeled the laser treatment in these figures.

Comment 7: The author should describe the statistics method in the Methods Section.

Reply 7: We have added the description of the statistics method to the Methods Section.

Added in the “Methods” section (Line 12, Page 25):

“**Statistical analysis.** All the data were presented as means ± standard deviation (SD). In order to test the significance of the observed differences between the study groups, analysis by variance (ANOVA) statistics was applied and a value of $P < 0.05$ was considered to be statistically significant.”

Reviewer #1 (Remarks to the Author):

The authors have satisfactorily addressed all my comments. Meanwhile, the reviewer also notice that the authors provided many new evidences and made sufficient amendments to address other reviewers' comments. As this research topic is extremely important and timely, I would like to recommendation publication of this work at its current form.

Reviewer #2 (Remarks to the Author):

The authors have properly address all my comments. I believe the manuscript is ready for publication.

Reviewer #3 (Remarks to the Author):

The author made the changes I requested and a good faith effort to improve the article. They also made more experiments to support their conclusions. I agree to accept this manuscript for publication.